# Death effector domain-containing protein induces vulnerability to cell cycle inhibition in triple-negative breast cancer

Yingjia Ni[1,2], Keon R. Schmidt[1,2], Barnes A. Werner[1,2], Jenna K. Koenig [1,2], Ian H. Guldner[1,2], Patricia M. Schnepp[1,2], Xuejuan Tan[1,2], Lan Jiang[1,2], Misha Host[1,2], Longhua Sun[1,2], Erin N. Howe[1,2], Junmin Wu[2,3], Laurie E. Littlepage [2,3,4], Harikrishna Nakshatri [4,5] & Siyuan Zhang [1,2,3,4]

Lacking targetable molecular drivers, triple-negative breast cancer (TNBC) is the most clinically challenging subtype of breast cancer. In this study, we reveal that Death Effector Domain-containing DNA-binding protein (*DEDD*), which is overexpressed in > 60% of TNBCs, drives a mitogen-independent G1/S cell cycle transition through cytoplasm localization. The gain of cytosolic *DEDD* enhances cyclin D1 expression by interacting with heat shock 71 kDa protein 8 (HSC70). Concurrently, *DEDD* interacts with Rb family proteins and promotes their proteasome-mediated degradation. *DEDD* overexpression renders TNBCs vulnerable to cell cycle inhibition. Patients with TNBC have been excluded from CDK 4/6 inhibitor clinical trials due to the perceived high frequency of Rb-loss in TNBCs. Interestingly, our study demonstrated that, irrespective of Rb status, TNBCs with *DEDD* overexpression exhibit a *DEDD*-dependent vulnerability to combinatorial treatment with CDK4/6 inhibitor and EGFR inhibitor in vitro and in vivo. Thus, our study provided a rationale for the clinical application of CDK4/6 inhibitor combinatorial regimens for patients with TNBC.

[1] Department of Biological Sciences, College of Science, University of Notre Dame, Notre Dame, IN 46556, USA. [2] Mike and Josie Harper Cancer Research Institute, University of Notre Dame, South Bend, IN 46617, USA. [3] Department of Chemistry and Biochemistry, College of Science, University of Notre Dame, Notre Dame, IN 46556, USA. [4] Indiana University Melvin and Bren Simon Cancer Center, Indianapolis, IN 46202, USA. [5] Departments of Surgery, Biochemistry and Molecular Biology, Indiana University School of Medicine, Indianapolis, IN 46202, USA. Correspondence and requests for materials should be addressed to S.Z. (email: szhang8@nd.edu)

Cancer is a highly heterogeneous genetic disease[1]. Despite the heterogeneous nature of the tumor, targeted anti-cancer therapies have achieved clinical success through the exploitation of functionally essential genomic alternations, or tumor-specific vulnerabilities[2,3]. In the past decade, a compendium of the conceptual frameworks explaining mechanisms of tumor vulnerabilities has been proposed and experimentally validated, including oncogene/non-oncogene addiction[4], synthetic lethality[5], collateral lethality[6], and synthetic essentiality[7]. A successful clinical translation of cancer genome studies (e.g., The Cancer Genome Atlas (TCGA) project)) for effective targeted anti-cancer regimens relies on two prerequisites: (1) tumor-specific genetic or epigenetic events that confer unique vulnerabilities (the Achilles' heel of cancer) to therapeutic targeting; (2) prognostic biomarkers coupled with an accessible clinical assay for selecting patients who will most likely respond to the targeted therapy.

Breast cancer targeted therapies are among the most successful examples of applying the concept of targeting tumor-specific vulnerabilities[8]. Estrogen receptor (ER)-targeting agents (e.g., tamoxifen) and human epidermal growth factor receptor 2 (HER2)-targeting therapeutics (e.g., trastuzumab) have become first-line treatments for ER-positive breast cancer and HER2-positive breast cancer, respectively. Extensive pre-clinical and clinical studies have demonstrated that biomarker-based (e.g., ER or HER2) patient selection for targeted therapies has led to a significant improvement in cancer treatment with reduced side effects compared to traditional chemotherapies[9]. Despite such clinical success, targeted therapy options for the most aggressive breast cancer subtype, triple-negative breast cancer (TNBC), remains limited. Because TNBC lack established therapeutic targets (e.g., ER and HER2), non-specific chemotherapy remains the primary treatment option for TNBC patients. TNBC exhibits a higher level of genome instability and a distinct mutational landscape[10,11]. Despite the highly heterogeneous nature of TNBC genome landscapes, some TNBC-specific tumor vulnerabilities are emerging as clinically translatable targets for TNBC patients. For example, poly-ADP ribose polymerase (PARP) was identified to have a synthetic lethal dependency on BRCA1 mutation and has been targeted in TNBC clinical trials, ultimately resulting in recent Food and Drug Administration (FDA) approval of PARP inhibitors for TNBC[12]. Additionally, bromodomain and extra-terminal (BET) protein[13] and the pyrimidine synthesis pathway[14] have shown promise to be targetable TNBC vulnerabilities.

In this study, we employed a pooled whole-genome RNA interference (RNAi) functional screen in TNBC cells to explore potential synthetic lethal pathways that synergize with epidermal growth factor receptor (EGFR)/HER2 inhibitor lapatinib (LAP).

Unexpectedly, we identified death effector domain-containing protein (DEDD) as a potential vulnerability from our screen. Interestingly, DEDD is significantly upregulated in >60% of TNBC tumors. While DEDD has been known to function as a pro-apoptotic protein in the nucleus[15], we found that DEDD is strongly expressed in the cytosol of tumor cells. Mechanistically, cytosolic DEDD promotes G1/S cell cycle transition through multiple mechanisms. First, DEDD interacts with heat-shock cognate 71 kDa protein (HSC70) to enhance cyclin D1 expression. Second, overexpressed cytosolic DEDD promotes the proteasome-mediated degradation of retinoblastoma (Rb) family proteins to enable G1/S transition. Addicted to an accelerated G1/S cell cycle progression, tumor cells with DEDD overexpression exhibit an increased susceptibility to the combinatorial treatment of cyclin-dependent kinases 4/6 (CDK4/6) and EGFR inhibitors. Furthermore, a combinatorial regimen of CDK4/6 and EGFR inhibitors synergistically inhibited the progression of TNBC xenografts and patient-derived xenograft (PDX) in vivo. These

pre-clinical results provide a strong rationale to extend recently FDA-approved CDK4/6 inhibitors to TNBC patients.

## Results

**DEDD upregulation confers a vulnerability to EGFR/HER2 inhibitor.** While TNBC tumors express EGFR, the clinical efficacy of anti-EGFR therapy in TNBC is low[16], suggesting the existence of alternative survival pathways that support TNBC proliferation under EGFR inhibition. Consistent with clinical observations, the proliferation of TNBC cells with high EGFR expression (Supplementary Fig. 1A) was not inhibited by EGFR/HER2 treatment (LAP) (Supplementary Fig. 1B) despite inhibition of phosphorylated (p)-EGFR, p-Akt, and p-Erk signaling (Supplementary Fig. 1C). Interestingly, although LAP treatment suppressed p-EGFR and downstream p-ERK, LAP did not effectively inhibit p-Akt at 24 h post treatment compared to 2 h of treatment (Supplementary Fig. 1C). This observation suggests that there is an alternative pathway that allows cells to adapt to the inhibition of the EGFR pathway. To identify such alternative pathways, we conducted a whole-genome loss-of-function RNAi screen by infecting the TNBC cell line (HCC1806; basal-like BL2 subtype) with DECIPHER Lentiviral shRNA Library Human Module 1 (5043 gene targets, 27,500 short hairpin RNAs (shRNAs)) followed by LAP treatment (Fig. 1a). We selected the top 200 ranked shRNA targets, which are decreased under the LAP treatment using the MAGeCK analysis software[17]. shRNA targets with reduced presentation under the LAP treatment ("drop-out" hits) were potentially critical for cell survival (Supplementary Data 1 and Supplementary Fig. 2A), particularly under EGFR/HER2 inhibition (Fig. 1a, b). To explore the clinical relevance of our screening result, we further examined gene alterations of the top 200 "drop-out" hits in breast cancer genome studies available at cBioPortal [http://www.cbioportal.org]. Among 200 hits, three genes (DEDD, ABCB10, and UAP1) are dysregulated (amplification and messenger RNA (mRNA) upregulation) in more than 40% of TNBC tumors (Fig. 1c) and in 20% of all breast tumors examined (Supplementary Fig. 2B, C). Interestingly, among the top dysregulated hits, a set of genes (DEDD, ABCB10, UAP1, KCNJ9, and PSAT1) are located close together on chromosome 1q23.3–42.1 (Fig. 1d) and are co-amplified in >15% of breast tumors from 164 cancer genome studies (Fig. 1e). Significantly, 67% of TNBC tumors have an upregulation of DEDD (Fig. 1c) as compared to a 35–43% dysregulation rate among all other breast cancer cases examined in METABRIC and the TCGA project (Supplementary Fig. 2B-D). Upregulation of DEDD, ABCB10, or UAP1 expression does not predict either overall or disease-free survival in TNBC patients who received current clinical treatment regimen (Supplementary Fig. 2E), suggesting that the genomic gain of 1q23.3–42.1, particularly DEDD, might have a unique role in conferring resistance to EGFR signaling inhibition. To independently validate our screening results, we used an independent set of shRNA to target DEDD, ABCB10, or UAP1 in multiple TNBC cell lines (Supplementary Fig. 3A, B and C). Multiple ABCB10 or UAP1 shRNA knockdowns only showed moderate effects with LAP treatment in HCC1806 cells (Supplementary Fig. 3D, E). Furthermore, knockdown of ABCB10 or UAP1 did not show a consistent resensitization effect on MDA-MB-468 cells to LAP treatment (Supplementary Fig. 3D, E). Compared to ABCB10 and UAP1, knocking down DEDD showed the most consistent and significant effect of sensitizing TNBC cells to the LAP treatment (Fig. 1f). Furthermore, we observed that knockdown of DEDD by DEDD siRNAs aborted sustained Akt phosphorylation at 24 h post treatment of LAP in TNBC cells (Supplementary Fig. 3F). Collectively, these results demonstrated that the abnormally high

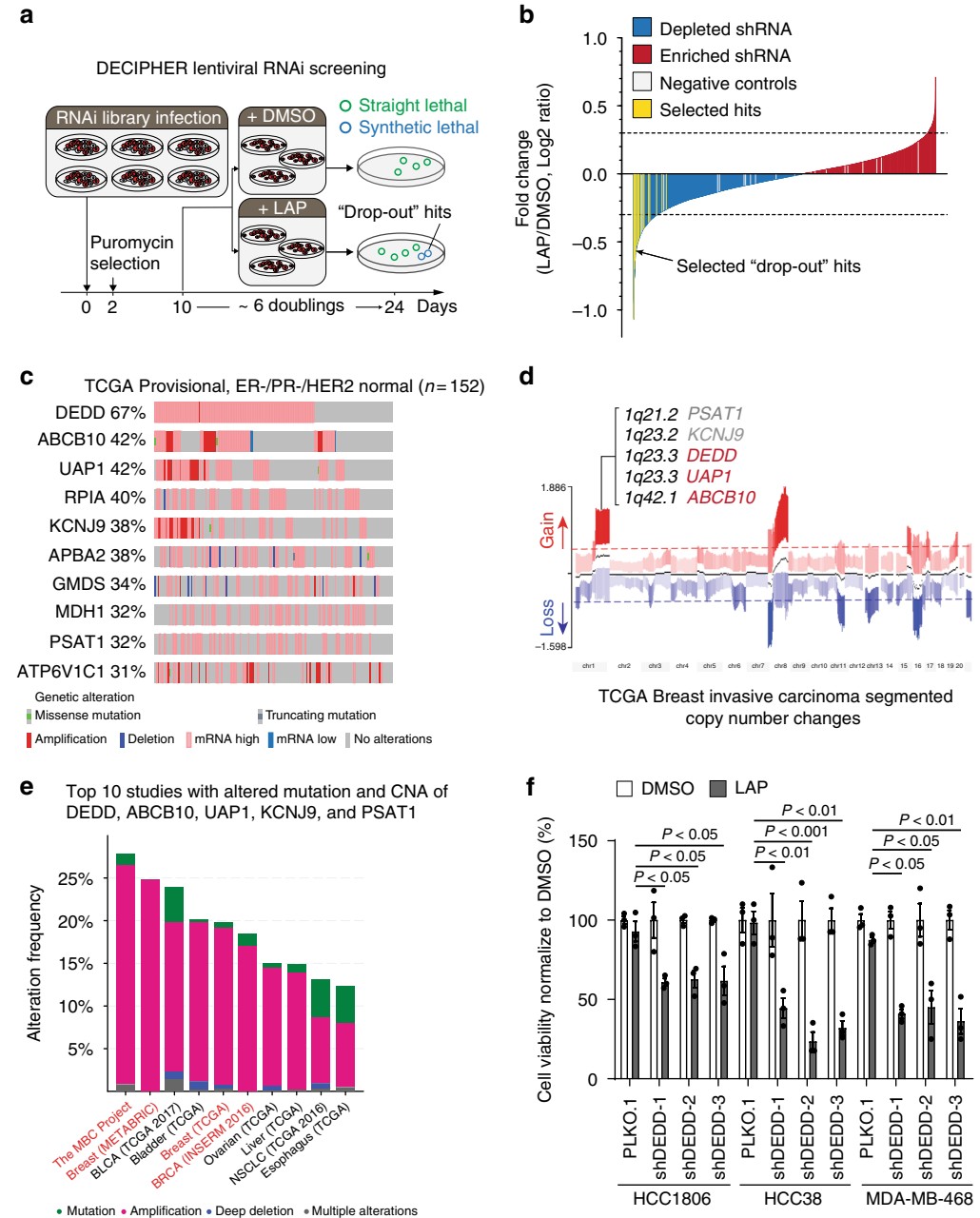

**Fig. 1** Death effector domain-containing DNA-binding protein (*DEDD*) upregulation confers a vulnerability to epidermal growth factor receptor/human epidermal growth factor receptor 2 (EGFR/HER2) inhibitor. **a** Schematic representation of the synthetic lethal bar-coded short hairpin RNA (shRNA) drop-out screen using triple-negative breast cancer (TNBC) cell line HCC1806 with the EGFR inhibitor lapatinib (LAP) treatment. **b** Bar-coded shRNAs enriched or depleted in the LAP-treated HCC1806 cells compared with dimethyl sulfoxide (DMSO)-treated cells. The bar-coded shRNAs of the top 200 depleted genes in LAP-treated cells were considered as drop-out hits (MAGeCK method, colored yellow). **c** Genetic and epigenetic alterations of the top 10 genes among top 200 depleted genes that are upregulated in estrogen-negative/progesterone receptor-negative/HER2 (ER−/PR−/HER2) normal tumors (*n* = 152) found within the Cancer Genome Atlas (TCGA). **d** Segmented view of the copy-number changes of *DEDD*, *ABCB10*, *UAP1*, *KCNJ9*, and *PSAT1* in TCGA breast-invasive carcinoma tumors. **e** Genome alteration frequency plot of top 10 cancer studies with *DEDD*, *ABCB10*, *UAP1*, *KCNJ9*, and *PSAT1* alterations across 164 studies in cBioPortal. **f** Cell counting assay validating knockdown of *DEDD* sensitizes TNBC cells to LAP treatment (error bars: means ± s.e.m.). Cells were normalized to DMSO control group in each shRNA or PLKO.1 (Control) group. All quantitative data were generated from a minimum of three replicates. *P* values were derived from one-way analysis of variance (ANOVA) with Dunnett's multiple comparison test comparing different shRNAs to the PLKO.1 group

expression of *DEDD* in TNBC confers resistance to anti-EGFR/HER2 treatment.

**High *DEDD* expression facilitates G1/S progression in TNBCs.** *DEDD* belongs to a large family of the death effector domain (DED)-containing proteins. Without known enzymatic activity,

*DEDD* executes its biological function primarily through protein–protein interactions via its DED domain[18]. Previous studies suggested that *DEDD* may interact with cyclin B1, decrease Cdk1/cyclin B1 activity, and regulate cell size during pre-mitosis phases by facilitating the G1-phase rRNA synthesis[19]. However, most studies have focused on the capacity of

*DEDD* to promote apoptosis through partnering with other DED-containing proteins[20]. Since *DEDD* is involved in pro-apoptotic processes, it is thought to have tumor suppressor activities[21]. Paradoxically, *DEDD* is aberrantly overexpressed in TNBC (Fig. 1c–e), implying that *DEDD* potentially has an essential role in TNBC development. To explore the TNBC-specific role of *DEDD*, we examined *DEDD* expression in seven TNBC cell lines (Fig. 2a). Knocking down *DEDD* led to a more significant inhibition of cell proliferation in TNBC cells with high expression of *DEDD* (HCC38 and MDA-MB-468) than TNBC cells with moderate *DEDD* expression (HCC1806) (Fig. 2b, c), suggesting that high *DEDD* expression TNBC cells are more dependent on *DEDD*-driven proliferative signaling. At the transcriptome level, knocking down *DEDD* globally downregulated EGFR-, AKT/MEK-, and cell cycle-associated gene signatures (Supplementary Fig. 4, RNA-sequencing

(RNA-seq) and gene set enrichment analysis (GSEA)). To examine the mechanisms by which *DEDD* overexpression impacts cell cycle regulation, we analyzed G1/S transition with a 5-ethynyl-2'-deoxyuridine (EdU) incorporation assay. We found that knocking down *DEDD* significantly decreased EdU incorporation in HCC38 and MDA-MB-468 TNBC cells (Fig. 2d, e), suggesting a role of *DEDD* in facilitating the G1 to S transition. To further investigate the relationship between *DEDD* and cell cycle progression in TNBC, we synchronized cell cycle progression by pretreating cells with microtubule inhibitor nocodazole (100 ng/mL) for 18 h, and then released cell cycle inhibition by replacing nocodazole media with a fresh culture medium. We found that *DEDD* protein peaked in cells at the G1 phase, but not other cell cycle phases (Supplementary Fig. 5A). Further time-course cell cycle analysis suggested that *DEDD* expression levels more closely correlated with cyclin D1

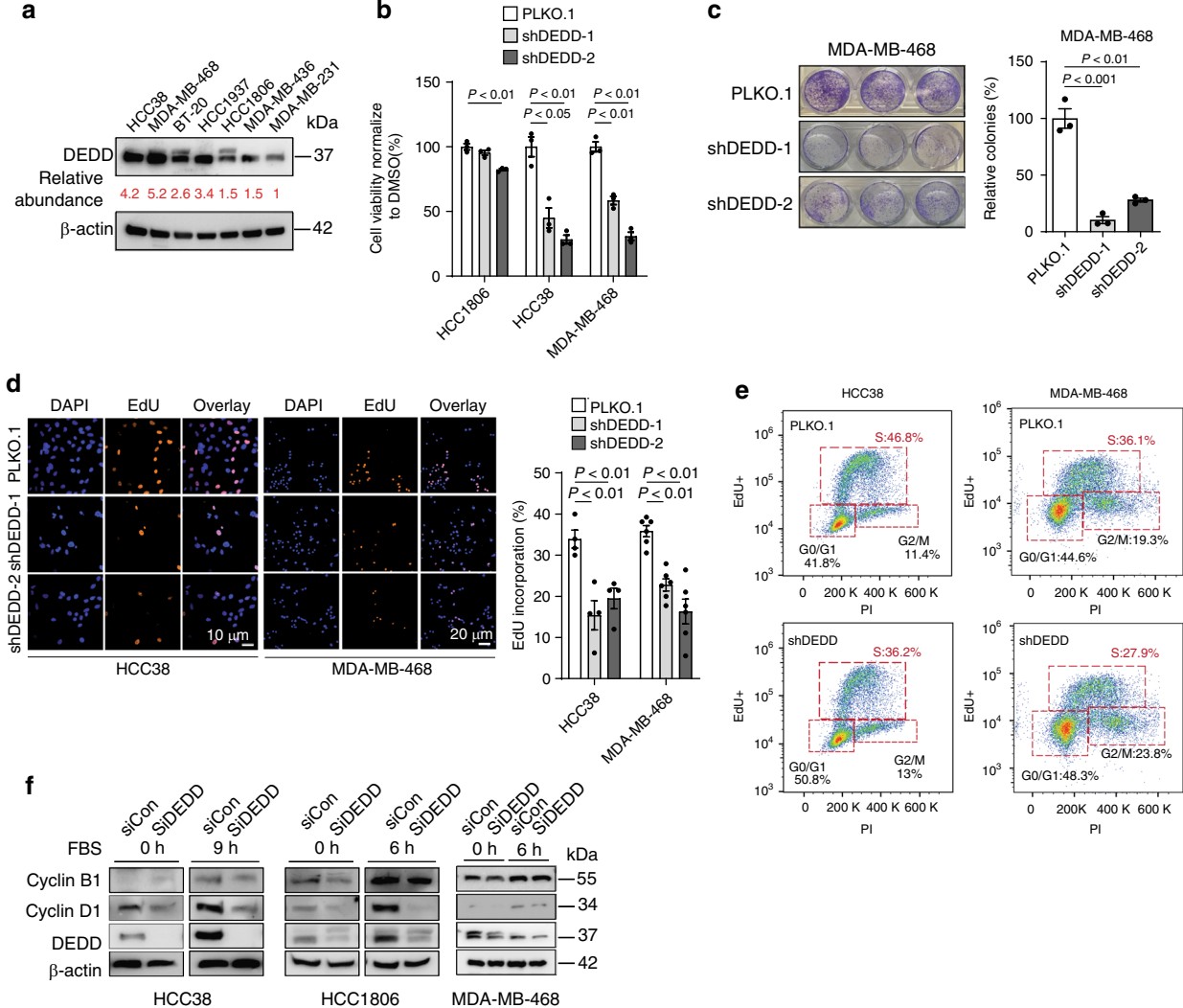

**Fig. 2** High death effector domain-containing DNA-binding protein (*DEDD*) expression facilitates G1/S progression in triple-negative breast cancers (TNBCs). **a** Western blots showing the endogenous *DEDD* expression across multiple TNBC cell lines. **b** Cell counting assay showing cell relative cell viability after *DEDD* knockdown in three TNBC cell lines. **c** Clonogenic assay showing relative colony number changes after *DEDD* knockdown in MDA-MB-468 cells. **d**, Left: Representative images showing 5-ethynyl-2'-deoxyuridine (EdU) incorporation in TNBC cells with or without short hairpin *DEDD* (sh*DEDD*) expression. 4',6-Diamidino-2-phenylindole (DAPI) staining (blue); EdU staining (orange). Right: Quantification of EdU incorporation. **e** Representative flow biaxial plots showing cell cycle population distribution in TNBC cells with or without sh*DEDD* expression. **f** Immunoblots showing cyclin B1 and cyclin D1 expression changes with or without small interfering *DEDD* (si*DEDD*) knockdown at 0 and 9/6 h after abolishing synchronization by fetal bovine serum (FBS) in HCC38/HCC1806/MDA-MB-468 cells. All quantitative data were generated from a minimum of three replicates. *P* values were derived from one-way analysis of variance (ANOVA) with Dunnett's multiple comparison test comparing different shRNAs to the PLKO.1 group. Error bars represent means ± s.e.m

expression (Supplementary Fig. 5B-D). Consistently across three TNBC cell lines, knocking down *DEDD* significantly decreased cyclin D1 expression, but not cyclin B1 expression (Fig. 2f), suggesting that *DEDD* facilitates G1/S cell cycle transition in TNBC through cyclin D1.

**Cytosolic DEDD is sufficient to facilitate G1/S transition**. *DEDD* nuclear localization leads to caspase activation, which induces apoptosis, and abolishing the nuclear translocation of *DEDD* decreases the apoptosis-promoting potential of the protein[15,20]. Interestingly, immunohistochemistry (IHC) staining for *DEDD* in TNBC tissue samples showed that *DEDD* primarily localizes to the cytoplasm of tumor cells that have high *DEDD* expression (Fig. 3a (left) and Supplementary Fig. 6A). Thus, we hypothesized that overexpression of *DEDD* in TNBC tumor cells has distinct tumor-promoting roles in the cytosol, which are independent of its known nuclear function. To study the cytosol-specific function of *DEDD* in promoting TNBC development, we constructed FLAG-tagged *DEDD* constructs with substitution mutations in all three previously identified nuclear localization sequences (ΔNLS) (Fig. 3a (right) and Supplementary Fig. 6B)[15]. Then, we investigated the functional impact of ectopic over-expression of wild-type (WT) *DEDD* or ΔNLS-*DEDD* in *DEDD*-diploid normal cell lines including 293FT and normal human breast epithelial cells (HMECs) (MCF-10A, KTB-21, and KBT-37)[22]. Ectopic overexpression of WT *DEDD* promoted G1–S transition in 293FT cells, which could be significantly diminished by preventing *DEDD* nuclear-cytosol shuttling using nuclear export inhibitor leptomycin (lepto) (Fig. 3b). This observation pointed toward an essential role of cytosolic *DEDD* in promoting G1/S transition. Moreover, the overexpression of cytosolic *DEDD* (FLAG-ΔNLS-*DEDD*) was sufficient to override the G1/S checkpoint and increase the proportion of cells in S phase inde-pendent of mitogen activation (serum starvation) as determined by increased EdU incorporation (Fig. 3c). Additionally, after abolishing synchronization with fetal bovine serum (FBS), there was no difference in relation to S-phase cell proportions between cells with *DEDD* overexpression and control 293FT cells (Sup-plementary Fig. 6C). We further examined *DEDD* expression in the recently established HMECs line. Compared to TNBC cell lines, HMECs generally have very low expression of endogenous *DEDD* (Supplementary Fig. 6D). Thus, HMECs provide an ideal system to study the functional role of *DEDD* in cell cycle reg-ulation. Overexpression of FLAG-ΔNLS-*DEDD* in KTB-21, KTB-37, and MCF-10A cells caused a prominent increase in the pro-portion of cells that passed through G1/S checkpoint 8 h after the cells were released from the nocodazole synchronization (Fig. 3d, e and Supplementary Fig. 6E). Cyclin profiling further revealed that cytosolic *DEDD* expression in KTB-37 cells led to an accel-erated cell cycle with induction of cyclin D1, A2, and E1 expression at 8 and 20 h post-nocodazole withdrawal (Fig. 3f). Interestingly, KTB cells with cytosolic *DEDD* overexpression appeared to be more sensitive to nocodazole-induced cell death (Supplementary Fig. 6F), and maintained an overall higher pro-liferation rate after nocodazole withdrawal (Supplementary Fig. 6F, G). Together, these data imply that abnormally high expression of cytosolic *DEDD* renders KTB cells more vulnerable to cell cycle-targeting agents.

**DEDD mediates HSC70-dependent G1/S progression and cyclin D1**. As an adaptor protein without known enzymatic activities, *DEDD* executes its functions through protein-protein interactions[23]. To explore interaction partners of *DEDD* in the cytosol, we immunoprecipitated the ΔNLS-*DEDD* protein com-plex and analyzed co-immunoprecipitated proteins by mass spectrometry. Interestingly, multiple peptides representing the HSC70 were identified (Supplementary Data 2). HSC70, also known as HSPA8, is a member of the heat-shock protein 70 families (HSP70). Independent co-immunoprecipitation assays further validated the interaction between *DEDD* and HSC70 in both normal *DEDD*-diploid 293FT cells and TNBC cells (Fig. 3g and Supplementary Fig. 6H). We next examined the necessity of HSC70 in mediating the ΔNLS-*DEDD*-dependent G1/S transi-tion. Importantly, knocking down HSC70 in two TNBC cell lines (Supplementary Fig. 6I) and one normal cell line KTB-37 (Fig. 3h) that ectopically overexpress ΔNLS-*DEDD* restored the S-phase cell cycle profiles to the levels seen in control cell lines without ΔNLS-*DEDD*. The reduced progression through G1/S checkpoint after HSC70 knockdown, even in the presence of ΔNLS-*DEDD*, suggests that HSC70 is an essential signaling partner for the cell cycle-promoting effect of *DEDD* in TNBCs. At the molecular level, despite mitogen deprivation, overexpression of ΔNLS-*DEDD* in KTB-34 and KTB-37 cells consistently increased the baseline levels of endogenous HSC70 protein (Fig. 3i). Also, knocking down HSC70 by small interfering RNA concurrently reduced *DEDD* protein levels, particularly in cells that overexpressed ΔNLS-*DEDD* (Fig. 3i). Moreover, the upre-gulated p-Akt/Erk signaling and cyclin D1 expression that was induced by ΔNLS-*DEDD* was inhibited by siHSC70 (Fig. 3i). Cyclin D1 is an activating regulatory subunit of CDK4/6, which are critical kinases driving G1/S transition[24]. Immuno-fluorescence staining confirmed increased cyclin D1 expression and a subsequent accelerated G1/S transition evidenced by increased EdU incorporation in ΔNLS-*DEDD*-expressed KTB cell lines (Fig. 3j). The above phenotypes imposed by overexpression of ΔNLS-*DEDD* were completely abolished by knocking down HSC70 (Fig. 3j and Supplementary Fig. 6I), suggesting that HSC70 is a downstream mediator of the cell cycle-promoting function of *DEDD*.

**Cytosolic DEDD promotes Rb/p107 proteasome degradation**. The tumor suppressor Rb is a major component of the G1/S checkpoint and blocks S-phase entry[24]. In normal cell cycle pro-gression, mitogen-induced cyclin D partners with the CDK4/6 complex to phosphorylate Rb in G1, resulting in Rb inactivation and de novo transcription that permits G1-to-S phase transition[25]. Unexpectedly, *DEDD* knockdown notably increased total Rb pro-tein levels in TNBC cells (Fig. 4a), while Rb mRNA levels either did not change significantly (HCC38) or decreased (HCC1806) (Fig. 4b). This suggests that there is post-translational regulation of Rb levels. Inhibition of protein translation by cycloheximide (CHX) supported reduced Rb protein turnover (or increased Rb stability) in *DEDD* knockdown TNBC cells at 6–12 h post treatment of CHX (Supplementary Fig. 7A). Notably, time-dependent inhibition of proteasome degradation machinery by MG-132 increased the total Rb protein levels and generated a prominent higher molecular weight smear in the control shRNA group, but not in cells with sh*DEDD* (Fig. 4c), suggesting that *DEDD* might mediate Rb post-translational modification via poly-ubiquitination and subsequent proteasome-mediated degradation.

Moreover, the progressive increase of the expression of either WT-*DEDD* or ΔNLS-*DEDD* induced a dose-dependent decrease of Rb protein levels, whereas MG-132 treatment stabilized the Rb protein levels in the *DEDD*-overexpressed cells (Fig. 4d). Similarly, in the normal HMEC KTB cells that were not treated with MG-132, the overexpression of ΔNLS-*DEDD* reduced Rb protein expression (Fig. 4e, MG-132, 0 h). Prolonged treatment with MG-132 (Fig. 4e, MG-132, 4–12 h) led to an accumulation of Rb as well as FLAG-ΔNLS-*DEDD*. Furthermore, overexpressed ΔNLS-*DEDD* co-immunoprecipitated with Rb (Fig. 4f), and

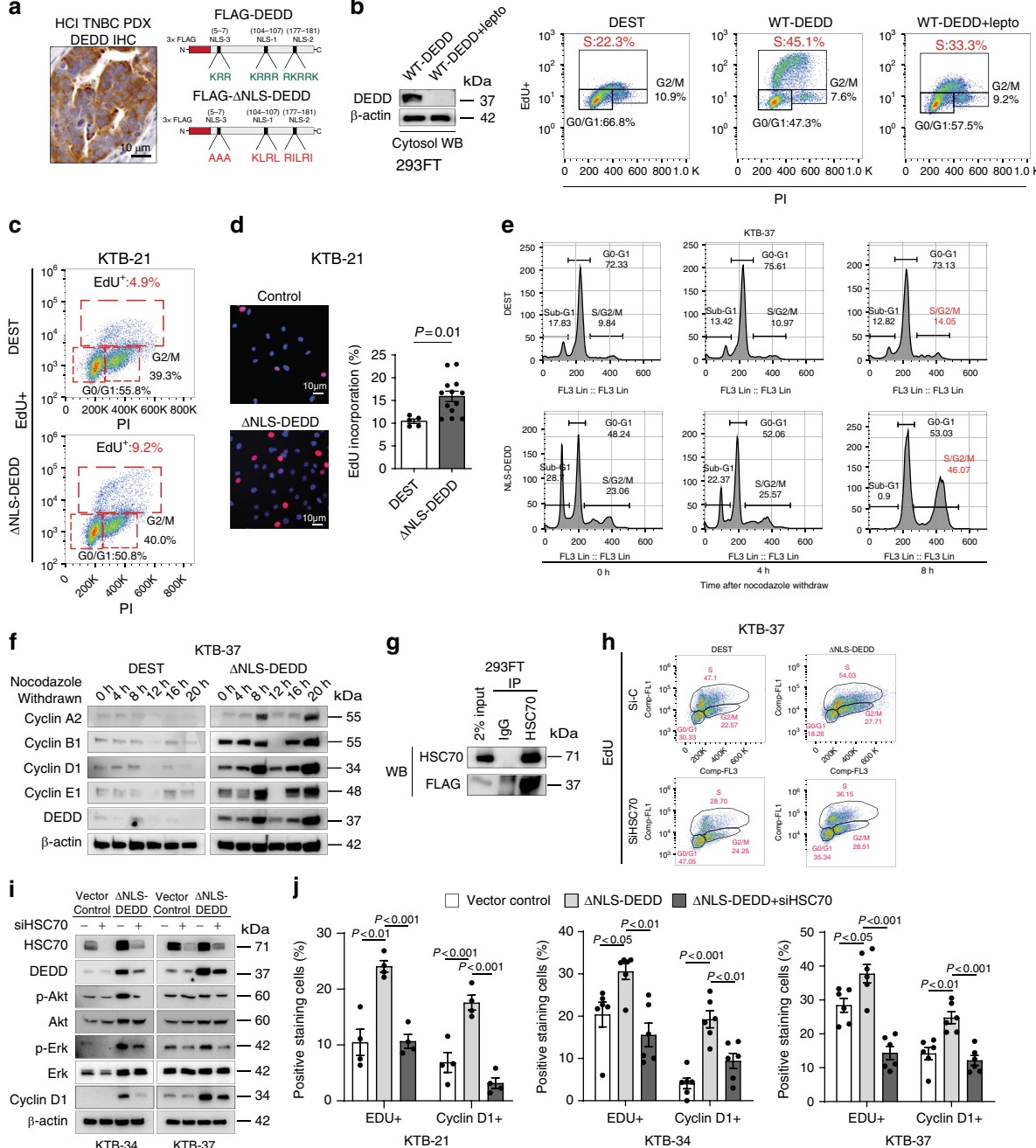

**Fig. 3** Death effector domain-containing DNA-binding protein (*DEDD*) mediates heat-shock cognate 71 kDa protein (HSC70)-dependent G1/S progression and cyclin D1. **a**, Left: immunohistochemistry (IHC) staining of *DEDD* in triple-negative breast cancer (TNBC) patient-derived xenograft (PDX) tissue. Right: Schematic of 3×-FLAG-tagged wild-type *DEDD* nuclear localization sequence (NLS) location and its mutated form. **b**, Left: Western blot showing FLAG-*DEDD* expression in 293FT cells with or without cytosol shuttle inhibitor leptomycin treatment (10 μg/mL, 3 h). Right: Representative flow cytometry biaxial plots of 293FT cells showing 5-ethynyl-2′-deoxyuridine (EdU) incorporation with or without expression of wild-type *DEDD* or cytosolic *DEDD* (ΔNLS-*DEDD*). **c** Representative flow cytometry biaxial plots of KTB-21 cells showing EdU incorporation with or without expression of cytosolic *DEDD* (ΔNLS-*DEDD*). DEST: control vector. **d**, Left: Representative immunofluorescence staining showing EdU incorporation of KTB cells with or without expression of cytosolic *DEDD*. 4′,6-Diamidino-2-phenylindole (DAPI): Blue; Red: EdU. Right: Quantification of EdU incorporation assay. **e** Representative flow cytometry biaxial plots of KTB-37 cells showing EdU incorporation with or without the expression of cytosolic *DEDD* after cell cycle release from nocodazole treatment (100 ng/mL) for 24 h. **f** Immunoblots showing cyclin profiles as well as *DEDD* expression of control plasmid or ΔNLS-*DEDD*-overexpressed normal mammary epithelial cells (KTB-37) at different time points after cells withdrawn from nocodazole treatment (100 ng/mL). **g** Immunoprecipitation assay showing the interaction between cytosolic *DEDD* and heat-shock cognate 71 kDa protein (HSC70) after overexpressing 3×-FLAG-DEL-NLS-*DEDD* plasmid in 293FT cells. **h** Representative flow cytometry biaxial plots showing EdU incorporation of HSC70-knockdown KTB-37 cells with or without the expression of cytosolic *DEDD*. **i** Western blots showing cell signaling changes of KTB cell lines with or without the treatment of HSC70 small interfering RNAs (siRNAs) for 2 days. **j** Bar chart showing the percentage of the immunofluorescence staining positive cells expressing either cytosolic *DEDD* or control vector in KTB cell lines with/without the treatment of HSC70 siRNAs for 2 days. Blue: DAPI; Red: EdU+. All quantitative data were generated from a minimum of three replicates. *P* values were derived from one-way analysis of variance (ANOVA) with multiple comparison test. Error bars represent means ± s.e.m.

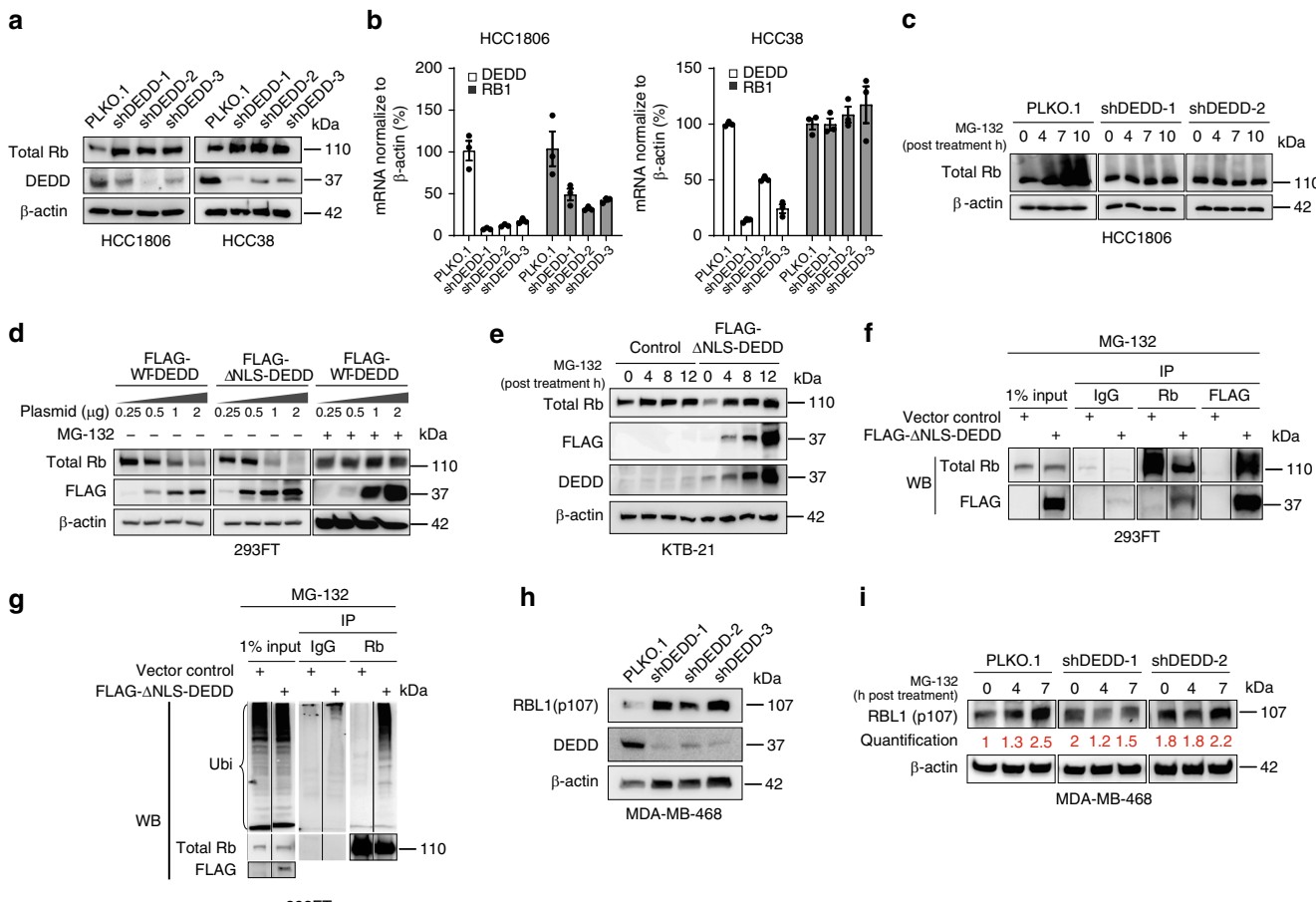

**Fig. 4** Cytosolic death effector domain-containing DNA-binding protein (*DEDD*) promotes retinoblastoma (Rb)/p107 proteasome degradation. **a** Western blots show Rb protein expression levels 6 h after abolishing synchronization by fetal bovine serum (FBS) in *DEDD* knockdown triple-negative breast cancer (TNBC) cell lines (HCC1806, HCC38, Rb intact). **b** Quantitative PCR showing both *DEDD* and Rb1 gene expression in *DEDD*-knockdown TNBC cell lines. **c** Western blots showing Rb protein expression in *DEDD*-knockdown HCC1806 cells under the MG-132 (proteasome inhibitor) treatment for indicated hours. **d** Western blots showing Rb protein expression levels in the 293FT cells treated with different dosages of 3×-FLAG-WT-*DEDD*- or 3×-FLAG-ΔNLS-*DEDD*-expressing plasmids with or without MG-132 treatment. **e** Western blots showing Rb protein expression levels in KTB-21 cells with or without the overexpression of cytosolic *DEDD* under the treatment of MG-132. **f** Immunoprecipitation assay showing the interaction between cytosolic *DEDD* and Rb protein after overexpressing 3×-FLAG-ΔNLS-*DEDD* in 293FT cells. **g** Immunoprecipitation assay showing Rb protein ubiquitination level after co-transfection of HA-ubiquitin and 3×-FLAG-ΔNLS-*DEDD* in 293FT cells. **h** Western blots showing Rb protein expression levels 6 h after abolishing synchronization by FBS in *DEDD*-knockdown TNBC cell lines (MDA-MB-468, Rb deficient). **i** Western blots showing Rb protein expression in *DEDD*-knockdown MDA-MB-468 cells under MG-132 treatment for indicated hours. All quantitative data were generated from a minimum of three replicates. *P* values were derived from one-way analysis of variance (ANOVA) with multiple comparison test. Error bars represent means ± s.e.m.

overexpression of ΔNLS-*DEDD* resulted in a significant increase in the poly-ubiquitination of Rb (Fig. 4g).

In addition to Rb protein, two other Rb family proteins, RBL1 (p107) and RBL2 (p130), were reported to be CDK4/6 substrates[26]. RBL1 serves as an alternative G1/S regulator in Rb-deficient cells. We observed that overexpression of RBL1, but not RBL2, occurred in 27% of TNBC tumors in TCGA dataset (Supplementary Fig. 7B). Due to the structural similarity between RB and RBL1, we further tested if *DEDD* also regulates RBL1 stability through ubiquitin-mediated degradation in the absence of Rb. In Rb-deficient MDA-MB-468 cells, knockdown of *DEDD* resulted in increased RBL1 (p107) expression, which phenocopied the Rb protein expression pattern in Rb-WT TNBC cell lines (Fig. 4h). Moreover, MG-132 treatment led to time-dependent (0–7 h) trend in the accumulation of RBL1, which was abolished by knocking down *DEDD* (Fig. 4i). We further validated that ectopically expressed cytosolic *DEDD* interacted with RBL1 and promoted its poly-ubiquitination, suggesting that *DEDD* regulates RBL1 in a manner similar to Rb (Supplementary

Fig. 7C and D). Collectively, the above observations point to a previously unknown function of cytosolic *DEDD* in promoting the degradation of Rb family G1 checkpoint proteins through ubiquitin-mediated proteasomal degradation, which subsequently drives a mitogen-independent progression through the G1/S transition.

**CDK4/6 inhibitor synergizes with EGFR inhibitor to suppress TNBCs.** Based on the multifaceted cell cycle-promoting mechanisms of *DEDD*, we hypothesized that gain-of-function *DEDD* in TNBC promotes mitogen-independent G1/S transition and confers addiction to an accelerated cell cycle program, thereby creating a tumor-specific vulnerability for therapeutic targeting. CDK4/6 inhibitors show significant clinical efficacy in cancer treatment by targeting the central G1/S transition machinery[27–29]. Three CDK4/6 inhibitors have recently received FDA approval for ER + breast cancer, including albociclib (PALBO) (Pfizer, FDA 2015), ribociclib (RIBO) (Novartis, FDA 2017), and abemaciclib (ABE) (Lilly, FDA 2017). Expression of

WT Rb, the perceived CDK4/6 downstream target, is currently considered a prerequisite for the clinical anti-cancer efficacy of CDK4/6 inhibitors[30].

Unfortunately, since Rb is frequently deleted in TNBC tumors[31], TNBC patients are excluded from CDK4/6 inhibitor clinical trials. Interestingly, we found that targeting CDK4/6 in combination with LAP in *DEDD*-overexpressed TNBC significantly and synergistically inhibited cell proliferation in both Rb-WT (HCC38, BT-20, HCC1937, HCC1806) and Rb-deficient (MDA-MB-468) TNBC cells (Fig. 5a and Supplementary Fig. 8A, B). This treatment phenocopied the synthetic lethality that occurred with the loss of *DEDD* in conjunction with LAP treatment (Fig. 1f). In combination with LAP, all three FDA-approved CDK4/6 inhibitors showed consistent synergy at fractional affected (FA) as low as 0.25 (Chou–Talalay model)[32] (Fig. 5b and Supplementary Fig. 8A, B). At the molecular level, treatment with LAP alone did not completely inhibit Akt phosphorylation 24 h post treatment (Supplementary Fig. 8C, and Fig. 5c). CDK4/6 inhibitor ABE alone consistently elevated Erk phosphorylation across the four TNBC cell line tested (Fig. 5c and Supplementary Fig. 8C). In contrast, the combination treatment of LAP and ABE (Combo) abolished both Erk and Akt signaling at 24 h post-Combo treatment (Fig. 5c and Supplementary Fig. 8C).

Forkhead box M1 (FoxM1) is a transcription factor and has been shown to be frequently upregulated in multiple cancers and linked to sustained Akt signaling[33,34]. Interestingly, we observed that FoxM1 expression was suppressed by ABE treatment in TNBC cell lines (Fig. 5c). We next asked whether gain-of-*DEDD* elicits vulnerability to Combo treatment. To determine if gain-of-cytoplasmic *DEDD* expression could lead to an acquired vulnerability to Combo treatment in normal cells, we treated control vector and ΔNLS-*DEDD*-overexpressed KTB cells with either single or Combo treatment (Fig. 5d). Ectopic expression of cytoplasmic *DEDD* (ΔNLS-*DEDD*) rendered the KTB cells resistant to LAP mono treatment, confirming the proliferation-driving function of cytoplasmic *DEDD*. Interestingly, KTB cells with overexpressed *DEDD* were more responsive to CDK4/6 inhibitor treatment and Combo treatment (Fig. 5d, right). At the molecular level, p-Akt expression is more sensitive to the combination treatment in the ΔNLS-*DEDD* KTB cell lines, determined by aberrant Akt signaling that is abolished by Combo treatment (Fig. 5e). This further validated that cytoplasmic *DEDD* (ΔNLS-*DEDD*) could further shift the sensitivity of TNBC cells to combo treatment. Overexpression of ΔNLS-*DEDD* in MDA-MB-436 cells, which moderately express *DEDD* (Fig. 2a), significantly enhanced sensitivity to the Combo treatment (Fig. 5f, Left), shifting the starting FA of the synergism zone (combination index (CI) <1) from FA = 0.5 in control vector-infected cells to FA = 0.24 in the ΔNLS-*DEDD*-expressed cells (Fig. 5f, right). The increased sensitivity to Combo treatment was also evident as a faster and more severe attenuation of p-Akt following Combo treatment in ΔNLS-*DEDD*-expressed cells (Fig. 5g). To test the long-term impact of Combo treatments on TNBC proliferation, we compared the therapeutic efficacy of Combo treatment to mono treatment using a clonogenic assay. Combo treatment significantly suppressed the colony forming capability in all three TNBC cells tested (Fig. 6a). Notably, Combo treatment reached close to 90% inhibition of proliferation in the Rb-deficient MDA-MB-468 cells (Fig. 6a).

To translate our findings to a treatment setting, we examined *DEDD* expression and localization as well as the sensitivity of TNBC cells to Combo treatment in vivo. First, we examined *DEDD* expression and localization in xenograft breast tumors derived from two TNBC cell lines (HCC1806, MDA-MB-468) and two TNBC PDXs (HCI-001, HCI-004) that were described

previously[35]. IHC analysis of *DEDD* showed heterogeneous but strong *DEDD* expression that was localized predominately to the cytoplasm of epithelial cells within the tumors (Fig. 6b, c). Next, we treated the TNBC xenograft and PDX models with Combo or mono treatment. In contrast to a single treatment, Combo treatment significantly decreased the size and the proliferative index (Ki-67 IHC staining) of the tumor, and also resulted in a moderate increase in apoptosis (cleaved caspase-3 IHC staining) (Fig. 6d–i). A similar response to Combo treatment was also observed in Rb-deficient MDA-MB-468 xenograft tumors (Supplementary Fig. 9A), evidenced by a moderate decrease of Ki-67 staining and a more significant increase of apoptosis indicated by cleaved caspase-3 staining in the residual tumors (Supplementary Fig. 9B,C). TNBC HCI-001 tumors were highly resistant to LAP treatment alone and moderately responsive to ABE (Fig. 6g), but sensitive to Combo treatment as evidenced by significantly reduced tumor size, reduced proliferation, and increased apoptosis (Fig. 6h, i).

Together, these results indicate that *DEDD* overexpression engages two mechanisms to achieve survival benefits for cancer cells. First, *DEDD* interacts with HSC70 to stabilize cyclin D1 expression. Second, *DEDD* promotes a poly-ubiquitin-dependent proteasome degradation of Rb family proteins, which facilitates G1/S transition independent of mitogen stimulation (Fig. 7). Furthermore, blocking both EGFR/HER2 signaling and cell cycle transition through the combinatory regimen of anti-EGFR treatment and CDK4/6 inhibitors successfully hindered TNBC tumor progression (Fig. 7).

## Discussion

Together, our results showed that amplification of chromosome 1q genes provides survival benefits for TNBC and can be targeted through CDK4/6 inhibitor-containing combinatorial therapies. Our study identifies *DEDD* as a regulator of mitogen-independent G1/S transition in TNBC tumor cells. Through interaction with HSC70, cytoplasmic *DEDD* induces cyclin D1 expression and promotes the proteasomal degradation of Rb family members (Fig. 7). More importantly, the gain-of-*DEDD* drives a previously unknown cancer-specific vulnerability to CDK4/6 targeting agents in combination with EGFR inhibitors, regardless of the Rb status of the TNBC cells (Figs. 5, 6 and Supplementary Fig. 8, 9).

It has been observed for more than a decade that breast tumors often exhibit chromosomal gains of both 1q and 8q (Fig. 1d). Specific genes on chromosome 8q (e.g., c-MYC) have been proven to drive tumorigenesis in a broad range of human cancers[36]. However, the potential chromosome 1q genes that bear functional significance in tumor initiation or development have not yet been fully identified. Chromosome 1q21 amplification is associated with hepatocellular carcinoma, while perplexingly, both 1q deletion and amplification were reported in breast cancers[37–39]. The above contradictory clinical observations suggests that a complex and tissue-context-dependent gene function resides in the chromosome 1q. Identifying such 1q genes and their associated signaling cascade is crucial for developing targeted therapies tailored towards tumors bearing 1q aberrations. Recently, Goh et al.[40] reported that overexpression of S100A7/8/9 genes harbored within 1q21.3 amplification confers a tumor-initiating cell phenotype in a subset of breast cancer cells. Serving as a biomarker, the presence of 1q21.3 amplification in cell-free DNA from patients' blood strongly correlates with early breast cancer recurrence[40]. Echoing the above study, we showed that the gain of 1q23.3 gene, *DEDD*, facilitates a mitogen-independent G1/S transition in TNBCs (Fig. 2d, e), the most aggressive subtype of breast cancer that frequently relapses. Although cytosol *DEDD* facilitates cell cycle transition (Fig. 3), overexpression of *DEDD*

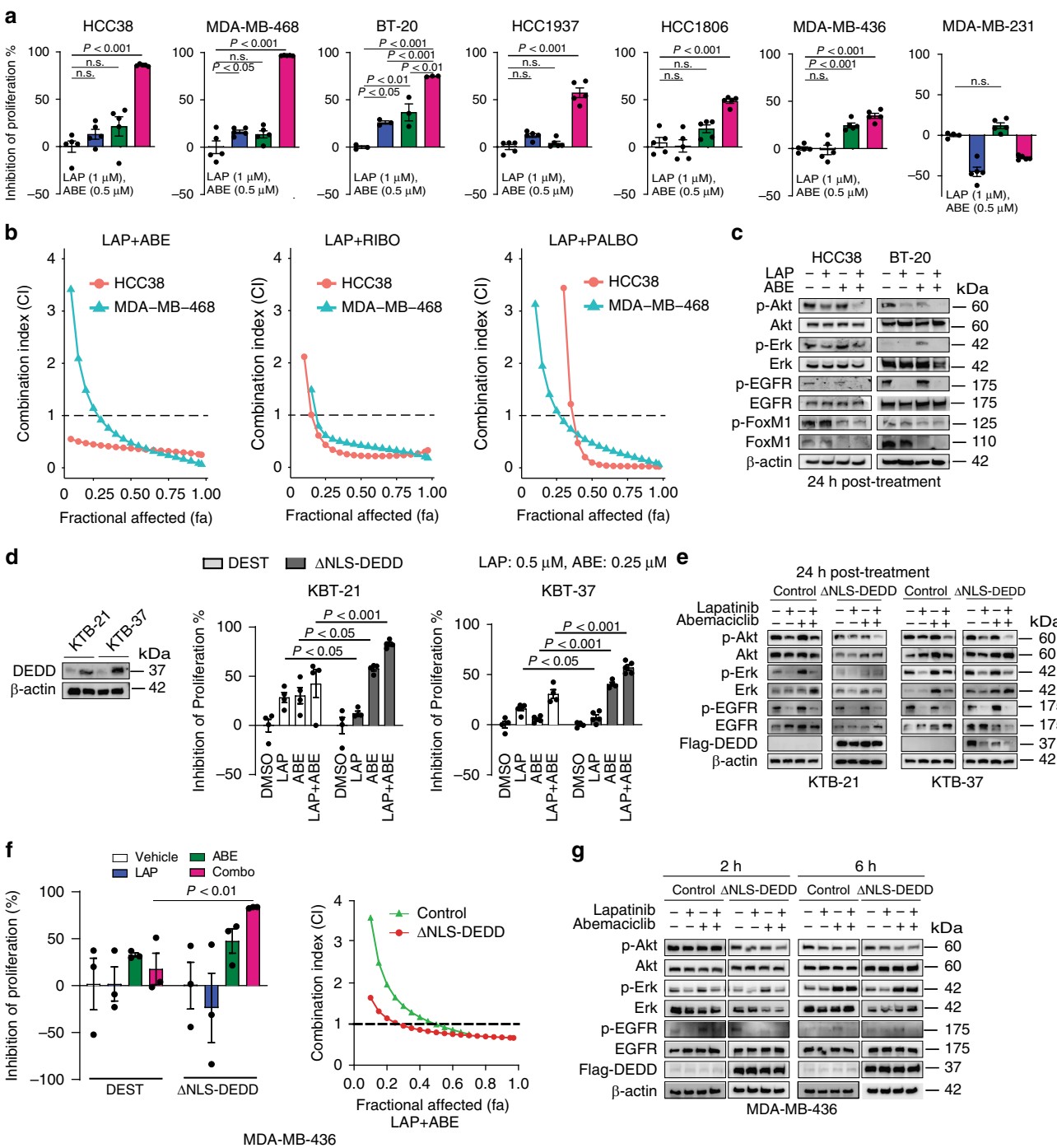

**Fig. 5** Cyclin-dependent kinases 4/6 (DK4/6) inhibitor synergizes with epidermal growth factor receptor (EGFR) inhibitor to suppress triple-negative breast cancer (TNBC) proliferation. **a** MTT (3-(4,5-dimethylthiazol-2-yl)-2,5-diphenyltetrazolium bromide) assay showing the inhibition of cell proliferation of TNBC cell lines treated with dimethyl sulfoxide (DMSO), lapatinib (LAP, 1 μM), abemaciclib (ABE, 0.5 μM), or Combo (LAP + ABE) for 48 h. **b** Drug synergism analysis showing synergism index on different FA (fraction affected) in TNBC cell lines. Dosage ratio: LAP:ABE = 4:1; LAP:RIBO (Ribociclib) = 2:1; LAP:PALBO (Palbociclib) = 1:2. **c** Western blots showing downstream signaling changes in TNBC cell lines after designated treatment for the indicated time. **d**, Left: Western blots showing cytosolic *DEDD* expression after lentiviral infected ΔNLS-*DEDD* in KTB cell lines. Right: MTT assay showing inhibition of proliferation of DEST- or ΔNLS-*DEDD*-expressing KTB cells treated with DMSO, LAP (0.5 μM), ABE (0.25 μM), or Combo for 48 h. **e** Western blots showing EGFR, Akt, and Erk signaling changes in DEST- or ΔNLS-*DEDD*-expressing KTB cells treated with DMSO, LAP (0.5 μM), ABE (0.25 μM), or Combo for 24 h. **f**, Left: MTT assay showing the inhibition of proliferation of MDA-MB-436 cells expressing or not expressing cytosolic *DEDD* after designated treatment. Cells were treated as in **d**. Right: Drug synergism analysis showing synergism index on different FA with the dosage ratio of 4:1 (LAP:ABE). **g** Western blots showing EGFR, Akt, and Erk signaling changes in MDA-MB-436 cells expressing or not expressing cytosolic *DEDD* treated with DMSO, LAP (1 μM), ABE (0.25 μM), or Combo for indicated hours. All quantitative data were generated from a minimum of three replicates. Error bars represent means ± s.e.m. For **a**, *P* values were derived from one-way analysis of variance (ANOVA) with multiple comparison test. For **d**, **f**, *P* values were derived from two-tailed *t* test. n.s., not significant

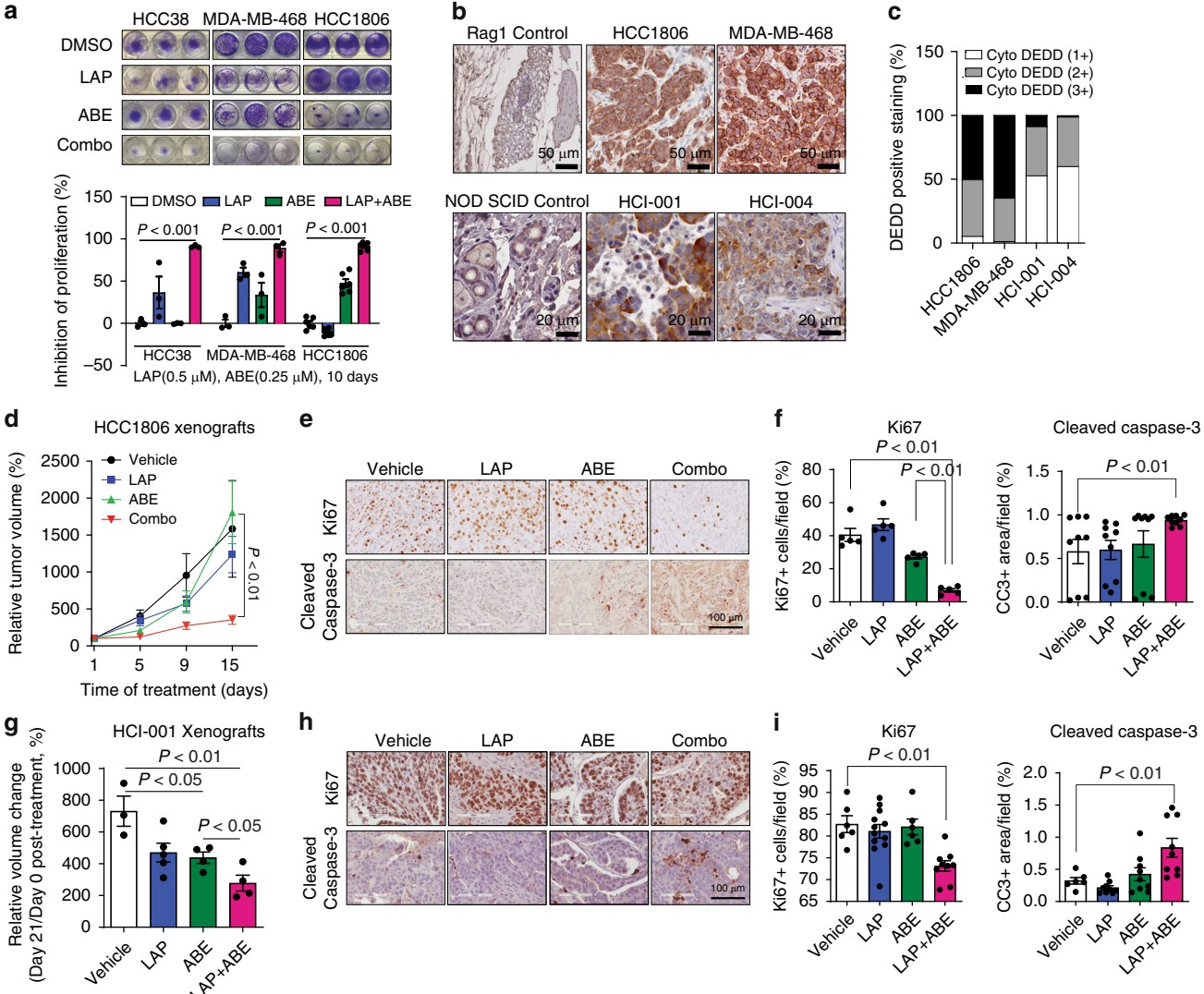

**Fig. 6** Cyclin-dependent kinases 4/6 (CDK4/6) inhibitor synergizes with epidermal growth factor receptor (EGFR) inhibitor to suppress triple-negative breast cancer (TNBC) tumor growth. **a**, Top: Representative pictures of clonogenic assay showing the inhibition of tumor progression in TNBC cell lines. Bottom: Quantification of the clonogenic assay. **b** Representative pictures of immunohistochemistry (IHC) staining showing death effector domain-containing DNA-binding protein (*DEDD*) expression and subcellular localization in TNBC xenografts and patient-derived xenograft (PDX) samples. **c** Quantification of *DEDD* expression from IHC staining. **d** Comparison of HCC1806 xenograft tumor volumes within mice treated with vehicle, lapatinib (LAP) (2 mg/per mouse per day), abemaciclib (ABE) (0.5 mg/per mouse per day) and the Combo treatment; $n > 10$ for each treatment group. **e** Representative pictures of IHC staining showing Ki-67 and cleaved caspase-3 expression in different treatment groups in HCC1806 xenografts. **f** Quantification of IHC staining in **e**. **g** Comparison of PDX tumor volumes from mice treated with vehicle, LAP (2 mg/per mouse per day), ABE (0.5 mg/per mouse per day), and the Combo; $n > 3$ for each treatment group. **h** Representative pictures of IHC staining showing Ki-67 and cleaved caspase-3 expression in different treatment groups in PDX. **i** Quantification of IHC staining in **h**. All quantitative data were generated from a minimum of three replicates. *P* values were derived from one-way analysis of variance (ANOVA) with multiple comparison. Error bars represent means ± s.e.m.

alone was not sufficient to drive normal mammary epithelial cell (KTB cells) transformation in soft-agar assays (data not shown). Likewise, *DEDD* did not influence overall survival and disease-free survival of TNBC patients who received traditional chemotherapies (Supplementary Fig. 2E). Thus, we deduce that *DEDD* may function as a facilitator of tumor progression, instead of a key driver in tumorigenesis.

Many proteins have distinct functions when translocated into different subcellular compartments. One of the well-studied examples is the tumor suppressor *PTEN*. In the cytosol, *PTEN* suppresses the phosphoinositide 3-kinase (PI3K)/AKT pathway through its lipid phosphatase activity, while a nuclear pool of *PTEN* maintains chromosomal stability[41]. *DEDD* has been well documented to be a nuclear-targeting protein through its three

nuclear localization sequences[18]. *DEDD* functions as an apoptosis facilitator in the nucleus by interacting with caspase-8 and caspase-10, and stimulating caspase-dependent cell death[15,20,21,42]. Surprisingly, once overexpressed in TNBC cells, *DEDD* primarily resides in the cytoplasm (Figs. 3a, 6b, and Supplementary Fig. 6A), suggesting that *DEDD* may play different roles in the regulation of cellular functions in the cytoplasm. In stark contrast to the apoptosis-promoting function of *DEDD* in the nucleus, our study revealed a role of cytosolic *DEDD* in promoting G1/S transition that provides a pro-proliferative benefit in both KTB and TNBC cells (Fig. 3d–f and Supplementary Fig. 6). It appears that the *DEDD* nuclear/cytosol equilibrium is critical for the cellular decision of cell proliferation or cell death. Previous studies have suggested that ubiquitination machinery

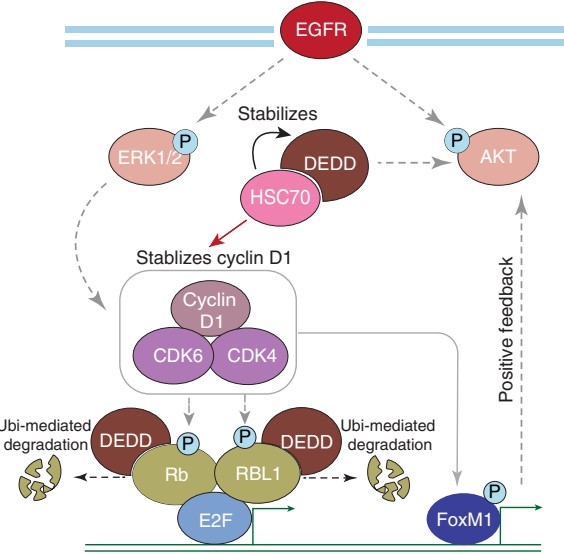

**Fig. 7** Proposed model of the roles of death effector domain-containing DNA-binding protein (*DEDD*) and combinatorial treatment mechanism in triple-negative breast cancer (TNBC). Proposed model of the roles of *DEDD* in TNBC and the combinatory therapy mechanism in *DEDD*-amplified TNBC

regulates protein nuclear/cytosol shuttling. For example, mono-ubiquitination of *PTEN* leads to its nuclear localization, whereas poly-ubiquitination of *PTEN* favors proteasomal degradation[43]. Similarly, in non-cancerous cells, *DEDD* exists in non-ubiquitinated, monoubiquitinated, and diubiquitinated forms. Mono-ubiquitin retains *DEDD* in the cytosol[20]. The mechanisms by which ubiquitination machinery fine-tunes the subcellular location of *DEDD* and its pro- or anti-proliferative functions in the TNBC cells warrants further study.

In our study, we demonstrated that cytosolic overexpression of *DEDD* facilitates G1/S transition (Fig. 3c–g and Supplementary Fig. 6). Since *DEDD* protein has not been shown to have enzymatic activity, it is intriguing to explore the mechanism by which *DEDD* regulates cell cycle transition. Our data provided the evidence that *DEDD* regulates G1/S cell cycle transition through multifaceted mechanisms. First, we demonstrated that the cell cycle-promoting function of *DEDD* is dependent on its cellular interacting partner HSC70 (Fig. 3g–j and Supplementary Fig. 6H, I). Previous studies have shown that *DEDD*, which lacks enzymatic activity, binds to DED-containing proteins, such as caspase-6, -8, and -10, through its DED domain[15,42], a protein–protein interaction domain only found in proteins that are involved in apoptosis signaling[44]. Besides the DED domain, no other functional domains have been identified on *DEDD*.

Interestingly, beyond the strictly DED-domain-mediated protein–protein interactions, our study revealed that cytosolic *DEDD* also binds to HSC70 chaperone protein in TNBC cells. This suggests that *DEDD* might engage a different set of protein binding partners to modulate cellular functions at different cell cycle stages. Our data show that *DEDD* and HSC70 interact to stabilize each other and cyclin D1, a critical component of the G1/S regulation complex cyclin D1–CDK4/6, which is known as one major binding substrate of HSC70 chaperone[45]. In addition, we also observed that other than cyclin D1, *DEDD* also mediates cyclin B1 expression in multiple TNBC cell lines (Fig. 2f), which is consistent with previous reports demonstrating that *DEDD* associates with cyclin B1 to regulate G2/M transition in mouse fibroblast cells[19]. Furthermore, it has been shown that *DEDD* associates with and stabilizes cyclin D3[46]. Thus, *DEDD* may play

multiple roles in cell cycle regulation other than G1/S transition in TNBC.

In this study, we also discovered that cytosolic *DEDD* interacts with Rb family proteins and modulates protein turnover through ubiquitin-mediated proteasomal degradation (Fig. 4g and Supplementary Fig. 7D). Previous studies of Rb regulation have been focused on its genetic deletion or mutation, as well as altered expression of the upstream regulators such as CDK4/6[25,26]. In contrast, the mechanisms that regulate Rb protein turnover are still largely understudied. One E3 ubiquitin ligase NRBE3 has been reported to be involved in the ubiquitin-dependent and proteasome-dependent degradation of Rb protein in vitro[47]. Another report suggests that p38 mitogen-activated protein kinase (MAPK) phosphorylates Rb on Ser567 in response to genotoxic stress. Hyperphosphorylation of Rb leads to its interaction with E3 ligases Hdm2[48]. However, the functional relevance of the above in relation to observations to tumorigenesis in vivo has not been addressed. In this study, we discovered that aberrant overexpression of *DEDD* greatly accelerated ubiquitination and proteasomal degradation of Rb family proteins, including Rb protein and Rb family protein RBL1 in TNBC cells (Fig. 4 and Supplementary Fig. 7). Mechanistically, considering its non-enzymatic nature, *DEDD* may act as an essential adaptor to engage the ubiquitin machinery. One possible mechanism could be that cytosolic *DEDD* directly recruits Rb and E3 ubiquitin ligases simultaneously into the same protein complex, thereby facilitating the poly-ubiquitination of Rb and subsequent proteasome degradation. Another possible mechanism could be that *DEDD* promotes p38 MAPK-mediated Rb hyperphosphorylation and indirectly enhances the ubiquitination of Rb. Exploring the above hypotheses in the TNBC context in future studies will add important mechanistic insight to the compendium of Rb regulation.

As discussed above, our study demonstrated that the major role of cytosolic *DEDD* is to promote G1/S transition. In *DEDD*-amplified TNBC cells, but not *DEDD* diploid cells, an accelerated G1/S transition occurs, resulting in a "non-oncogene addiction phenotype." These cells are particularly sensitive to cell cycle disruption by either knocking down *DEDD* or inhibiting CDK4/6 (Figs. 2b–e, 5a, b and Supplementary Fig. 8). Importantly, while inhibition of CDK4/6 alone only suppresses tumor cell proliferation to some degree in multiple TNBC cell lines (Figs. 5a, 6a–g and Supplementary Figs. 8, 9), targeting the EGFR pathway synergizes with CDK4/6 inhibitors (abemaciclib, palbociclib, ribomaciclib), leading to a synthetic lethal effect in both Rb-WT and Rb-deficient TNBC cells. The combination treatment of both EGFR and CDK4/6 inhibitors effectively decreases Akt phosphorylation compared to EGFR inhibitor alone in responding TNBC cell lines (Fig. 5c–g). Although LAP is effective in inhibiting p-Akt transiently (Supplementary Fig. 1C), TNBCs quickly adopt a mechanism that consistently sustains Akt signaling, and this sustained p-Akt is partially *DEDD* dependent (Fig. 5c and Supplementary Figs. 1D, 3F, 8C). It has been shown that female mice lacking *DEDD* are infertile owing to unsuccessful decidualization, which appears to be caused by decreased Akt levels, since polyploidization was restored in *DEDD*-deficient decidual cells by overexpression of Akt[49]. Thus, *DEDD* is highly involved in maintaining Akt signaling, which is different from the cell death signaling that has previously been well investigated. Interestingly, our data demonstrated that further inhibiting CDK4/6 activity effectively abolished p-Akt (Fig. 5c and Supplementary Fig. 8C). This observation suggests that an intriguing cross-talk between the CDK4/6 cell cycle cascade and Akt signaling exists, which is consistent with a recent report showing synergy between CDK4 and HER2 inhibition in HER2 + breast tumor model[50]. Mechanistically, we envision two potential mechanisms

contributing to the observed synergy. First, previous studies have demonstrated that cyclin D1–CDK4/6 complex down-regulates TSC1/2 and promotes mTORC1 activation[51]. Since it has been well established that the mTORC1 activation leads to PI3K and MAPK inhibition through a negative feedback loop, upregulation of cyclin D1 by overexpression of *DEDD* (Fig. 3f–j) will likely lead to a hyper-activation of mTORC1, rendering an attenuated PI3K/MAPK pathway activation[49]. The inhibition of cyclin D1–CDK4/6 complex by CDK4/6 inhibitors phenocopies mTORC1 inhibition and unleashes the activity of the MAPK pathway. Indeed, we observed that CDK4/6 inhibitor treatment alone increased the ERK phosphorylation in multiple TNBC cells that overexpress *DEDD* (Fig. 5c and Supplementary Fig. 8C). Adding LAP to CDK4/6 inhibition completely abolished this negative feedback activation of ERK 24 h post treatment (Fig. 5c and Supplementary Fig. 8C). Second, in addition to Rb and TSC1/2, transcription factor forkhead box M1 (FoxM1) has been identified as a CDK4/6 substrate through a proteomics screen[52]. FoxM1 is a typical proliferation-associated transcription factor that stimulates proliferation by promoting S-phase and M-phase entry, and is additionally involved in the proper execution of mitosis[33]. Previous reports demonstrated that FoxM1 activates the Akt pathway via c-Met and causes resistance to tyrosine kinase inhibitors[53]. Other literature shows platelet-derived growth factor subunit a (PDGFA) is a transcription target of FoxM1, and FoxM1 promotes breast cancer tumorigenesis by transcriptionally activating PDGFA, which led to further Akt phosphorylation in breast cancer cells[54]. Supporting such findings, we observed that phosphorylation and total expression of FoxM1 were effectively blocked by CDK4/6 inhibitors (Fig. 5c), which can explain the synergistic effect of applying both EGFR and CDK4/6 inhibitors in TNBC. Future investigation of *DEDD*-mediated FoxM1-Akt cross-talk will further guide rationally designed combinatorial therapy for TNBC patients.

In summary, our study revealed a mechanism governing cell cycle progression. While providing proliferative benefit, the gain of chromosome 1q gene *DEDD* also confers a *DEDD*-dependent vulnerability to CDK4/6 targeting combinatorial therapies. As CDK4/6 inhibitors are widely used in the clinic, our study provided a crucial pre-clinical rationale for a readily clinically translatable combinatorial strategy for TNBC patients.

## Methods

**Chemicals and cell culture.** LAP (L-4899) and palbociclib (P-7766) were purchased from LC Laboratories. Abemaciclib (HY-16297), ribociclib (HY-15777), and nocodazole (HY-13520) were purchased from MedchemExpress Inc. Leptomycin B (9676S) was purchased from Cell Signaling Technology. MG-132 (80053-196) was purchased from VWR. CHX (01810) was purchased from Sigma-Aldrich. The 293FT (R70007) cells were purchased from Thermo Fisher Scientific Company. BT-20, BT-474, HCC1806, HCC1937, HCC38, MDA-MB-231, MDA-MB-436, and MDA-MB-468 cell lines were obtained from ATCC (Manassas, VA, USA). The 293FT cell line was cultured in Dulbecco's modified Eagle's medium (DMEM) high glucose supplemented with 10% FBS and 1% penicillin–streptomycin (Pen-Strep). BT-20, BT-474, MDA-MB-231, MDA-MB-436, and MDA-MB-468 cell lines were cultured in DMEM/Ham's F-12 (1:1) supplemented with 10% FBS and 1% Pen-Strep. HCC1806, HCC1937, and HCC38 cell lines were cultured in RPMI-1640 medium supplemented with 10% FBS and 1% Pen-Strep. The MCF-10A was originally purchased from ATCC and cultured in DMEM/F-12 medium supplemented with 5% horse serum (Invitrogen), 20 ng/mL EGF, 10 μg/mL insulin, 0.5 μg/mL hydrocortisone, 100 ng/mL cholera toxin, and 1% Pen-Strep. Normal human mammary epithelial lines from Komen Tissue Bank (KTB) were obtained from our collaborator Dr. Nakshatri and cultured in DMEM/F-12 supplemented with 10% FBS, 0.4 μg/mL hydrocortisone, 5 μg/mL insulin, 20 ng/mL EGF, 24 mg/L adenine, and 5 μM ROCK (Rho-associated, coiled-coil containing protein kinase) inhibitor (Y-27632). All cell lines were maintained at 37 °C in a 5% $CO_2$ humidified environment and subcultured upon reaching approximately 90% confluence. Twenty-four hours before experiments, cells were re-plated into new Petri dishes with fresh medium at about 50% confluency. BT-20, BT-474, HCC1806, HCC1937, HCC38, MDA-MB-231, MDA-MB-436, MDA-MB-468, and MCF-10A were authenticated by DNA analysis by Genetica Cell Line Testing.

**Lentiviral transfection assay and plasmids.** Human *DEDD* complementary DNA (cDNA) sequence (refGene_NM_001039711 range = chr1:161,122,147–161,124,462) It was obtained from the UCSC Genome Browser.

*DEDD* coding sequence:
5′-ATGGCGGGCCTAAAGCGGCGGGCAAGCCAGGTGTGGCCAGAAGA GCATGGTGAGCAGGAACATGGGCTGTACAGCCTGCACCGCATGTTTGAC ATCGTGGGCACTCATCTGACACACAGAGATGTGCGCGTGCTTTCTTTCCT CTTTGTTGATGTCATTGATGACCACGAGCGTGGACTCATCCGAAATGGA CGTGACTTCTTATTGGCACTGGAGCGCCAGGGCCGCTGTGATGAAAGTA ACTTTCGCCAGGTGCTGCAGCTGCTGCGCATCATCACTCGCCACGACCT GCTGCCCTACGTCACCCTCAAGAGGAGACGGGCTGTGTGCCCTGATCTT GTAGACAAGTATCTGGAGGAGACATCAATTCGCTATGTGACCCCCAGAG CCCTCAGTGATCCAGAACCAAGGCCTCCCCAGCCCTCTAAAACAGTGCC TCCCCACTATCCTGTGGTGTGTTGCCCCACTTCGGGTCCTCAGATGTGTA GCAAGCGGCCAGCCCGAGGGAGAGCCACACTTGGGAGCCAGCGAAAAC GCCGGAAGTCAGTGACACCAGATCCCAAGGAGAAGCAGACATGTGACA TCAGACTGCGGGTTCGGGCTGAATACTGCCAGCATGAGACTGCTCTGCA GGGCAATGTCTTCTCTAACAAGCAGGACCCACTTGAGCGCCAGTTTGAG CGCTTTAACCAGGCCAACACCATCCTCAAGTCCCGGGACCTGGGCTCCA TCATCTGTGACATCAAGTTCTCTGAGCTCACCTACCTCGATGCATTCTGG CGTGACTACATCAATGGCTCTTTATTAGAGGCACTTAAAGGTGTCTTCAT CACAGACTCCCTCAAGCAAGCTGTGGGCCATGAAGCCATCAAGCTGCTG GTAAATGTAGACGAGGAGGACTATGAGCTGGGCCGACAGAAACTCCTGA GGAACTTGATGCTGCAAGCATTGCCCTGA-3′.

*DEDD* DEL-NLS coding sequence:
5′-ATGGCGGGGCCTAGCGGCAGCAGCAAGCCAGGTGTGGCCAGAAGA GCATGGTGAGCAGGAACATGGGCTGTACAGCCTGCACCGCATGTTTGAC ATCGTGGGCACTCATCTGACACACAGAGATGTGCGCGTGCTTTCTTTCC TCTTTGTTGATGTCATTGATGACCACGAGCGTGGACTCATCCGAAATGG ACGTGACTTCTTATTGGCACTGGAGCGCCAGGGCCGCTGTGATGAAAGT AACTTTCGCCAGGTGCTGCAGCTGCTGCGCATCATCACTCGCCACGACC TGCTGCCCTACGTCACCCTCAAGCTGAGACTGCTGTGTGCCCTGATCT TGTAGACAAGTATCTGGAGGAGACATCAATTCGCTATGTGACCCCCAGA GCCCTCAGTGATCCAGAACCAAGGCCTCCCCAGCCCTCTAAAACAGTGC CTCCCCACTATCCTGTGGTGTGTTGCCCCACTTCGGGTCCTCAGATGTGT AGCAAGCGGCCAGCCCGAGGGAGAGCCACACTTGGGAGCCAGCGAATC CTGCGGGATCTCAGTGACACCAGATCCCAAGGAGAAGCAGACATGTGACA TCAGACTGCGGGTTCGGGCTGAATACTGCCAGCATGAGACTGCTCTGCA GGGCAATGTCTTCTCTAACAAGCAGGACCCACTTGAGCGCCAGTTTGAG CGCTTTAACCAGGCCAACACCATCCTCAAGTCCCGGGACCTGGGCTCCA TCATCTGTGACATCAAGTTCTCTGAGCTCACCTACCTCGATGCATTCTGG CGTGACTACATCAATGGCTCTTTATTAGAGGCACTTAAAGGTGTCTTCAT CACAGACTCCCTCAAGCAAGCTGTGGGCCATGAAGCCATCAAGCTGCTG GTAAATGTAGACGAGGAGGACTATGAGCTGGGCCGACAGAAACTCCTGA GGAACTTGATGCTGCAAGCATTGCCCTGA-3′.

Human *DEDD* cDNA coding sequence was cloned into lentivirus mammalian expression vector pLOVE empty (Addgene #15948) or pLenti-CMV-puro-DEST (Addgene #17452) with an N-terminal 3×-FLAG tag. *DEDD* mutant entry vector was assembled through GeneArt Gene Synthesis (Thermo Fisher Scientific). For lentiviral production, a lentiviral expression vector was co-transfected with second-generation lentivirus packing vectors psPAX2 and pMD2.G (Addgene) into 293FT cells using CalFectin (SignaGen, Rockville, MD, USA). Forty-eight hours after transfection, breast cancer cell lines were stably infected with viral particles. pLOVE-GFP served as packaging and infection efficiency control. Lentiviral-based pLKO.1 *DEDD* shRNA vectors were purchased from Sigma-Aldrich Co. (St. Louis, MO, USA). The control shRNA vector (pLKO.1 scrambled shRNA) was from Addgene Inc. (Cambridge, MA, USA).

The *DEDD*-targeting shRNA sequences used in the lentiviral constructs were: sh*DEDD*−493, 5′-CCGGGAGGAGGAGCATCAATTCGCTATCTCGAGATAGCGA ATTGATGTCTCCTCTTTTTTG-3′ (targeting CDs); sh*DEDD*-867, 5′-CCGGCA TCATCTGTGACATCAAGTTCTCGAGAACTTGATGTCACAGATGATGTTTT TTG-3′ (targeting CDs); sh*DEDD*1056, 5′-CCGGGAAACTCCTGAGGAACTTG ATCTCGAGATCAAGTTCCTCAGAGTTTCTTTTTTG-3′.

Stable expression or knockdown clones were selected by culturing cells with puromycin (1–5 μg/mL) for at least 48 h. The uninfected cells were also treated with puromycin as a control. Infected cells have a multiplicity of infection (MOI) around 0.3–0.6.

**Primers.** Primers for constructing 3×-FLAG-*DEDD* or 3×-FLAG-Del-NLS-*DEDD*: N-3×-FLAG-*DEDD*—forward primer: melting temperature (TM): 72.4 ℃ 5′-CACCATG GAC TAC AAA GAC CAC GAC GGC GAC TAC AAA GAC CAC GAC ATC GAC TAC AAA GAC GAT GAC GAC AAGATGGCGGGCCTA. N-3×-FLAG-*DEDD*—reverse primer: TM: 72.1 ℃ 5′-TCA GGG CAA TGC TTG CAG CAT CAA GTT CCT CAG GAG TTT CTG TCG GCC CAG CTC ATA GTC. Primers for detecting gene mRNA level: *DEDD*—forward primer: TM: 56.6 ℃ 5′-ATGGACGTGACTTCTTATTG-3′. *DEDD*—reverse primer: TM: 56.4 ℃ 5′-ATACTTGTCTACAAGATCAGGG-3′.

RB1—forward primer: TM: 57.2 ℃
5′-CAGAAATGACTTCTACTCGAAC-3′.
RB1—reverse primer: TM: 58.9 ℃
5′-AATGTGGCCATAAACAGAAC-3′.
ABCB10—forward primer: TM: 62.1 ℃
5′-CCATATTTCAGGATTTCAGCC-3′.
ABCB10—reverse primer: TM: 54.8 ℃
5′-GACTAATAGTTCCAGAAGCAG-3′.
UAP1—forward primer: TM: 56.7 ℃
5′-GATAGTCAGAATGGGAAAGAC-3′.
UAP1—reverse primer: TM: 56.3℃
5′-GCATAGGAGATAAGAGGAGAG-3′.

**Genome-wide bar-coded shRNA screening and MAGeCK analysis.** In vitro pooled drop-out screening by infect HCC1806 cells at 0.3 MOI with DECIPHER shRNA lentiviral library (Cellecta Inc., human model 1, Cat. #DHPAC-M1-P), which includes about 27,000 shRNA constructs targeting 5043 genes on human signaling pathways. After 8 days of puromycin (1 μg/mL) selection, infected cells were treated with either dimethyl sulfoxide (DMSO) or LAP (2 μM) for 14 days with replacing fresh medium containing drug every 48 h. After 14 days of treatment, the cells are collected and sent for barcodes sequencing by Illumina sequencing. The row barcode counts were analyzed for the "drop-out" hits using the MAGeCK software as described before[17].

**Cell proliferation assays.** For MTT (3-(4,5-dimethylthiazol-2-yl)-2,5-diphenyltetrazolium bromide) assays, cells (3000/well) were seeded in at least triplicate in 96-well plates and treated with various concentrations of a single treatment or combination treatment for designated times. Cell proliferation was determined by incubating with MTT for 3 h and then in lysing buffer (50% DMF (N, N-dimethyl formamide) (Sigma, D-4254), 10% SDS (sodium dodecyl sulfate) (Sigma, L-4509)) overnight. The 96-well plate was read at 570 nm absorbance by a microplate reader. We used the wells with cells untreated as 100% control and the wells with only medium as 0% control. Percentage of inhibition of cell proliferation was calculated as $[100 - ((OD\ treated\ - OD\ empty)/(OD\ untreated - OD\ empty) \times 100]$. For direct cell counting, cells (3000/well) were seeded in at least triplicate in 96-well plates and treated with various concentrations of the single treatment or combination treatment for designated times. Cell proliferation was determined by directly counting cell numbers in each well using hemacytometer and compared using the wells with untreated cells as 100% control and the wells with only medium as 0% control. For the clonogenic assay, cells were plated approximately 3000 cells/well in 24-well plates in 10% FBS. After 24 h, cells were treated in triplicate with various concentrations of a single or combination treatment for approximately 2 weeks with media replacement and treatment every 48 h. Cells were washed with phosphate-buffered saline (PBS), fixed, and stained with a solution containing 6% glutaraldehyde and 0.5% crystal violet. Plates were dried, and each well was photographed and quantified using ImageJ 1.43u (National Institutes of Health, Bethesda, MD, USA).

**Quantitative RT-PCR assay.** Total cell RNA will be extracted using TriZol reagent (Invitrogen) following the manufacturer's instructions. One microgram of total RNA was subjected to reverse transcription to synthesize cDNA using the Verso cDNA Kit (Thermo Fisher Scientific). A 20 μL volume reaction consisted of 2 μL reverse transcription product and 20 nM of both forward and reverse primer. The samples were incubated with 2× iTaq™ Universal SYBR® Green Supermix (Bio-Rad). Relative expression levels were calculated using the $2^{-\Delta\Delta CT}$ method.

**EdU incorporation assay.** Twenty-four hours before the experiment, the culture medium was replaced with medium without FBS. This would synchronize the cells in G0. After 24-h serum starvation, the cells were treated with 1 ng/mL EGFs to induce cell cycle reentry. Cells were treated with 10 μM of EdU for 1–6 h. Cells were collected and processed following the protocol of either Click-iT® Plus EdU Alexa Fluor® 488 Flow Cytometry Assay Kit or by Click-iT™ EdU Alexa Fluor™ 594 Imaging Assay. In the Click-iT® Plus EdU Alexa Fluor® 488 Flow Cytometry Assay Kit (Thermo Fisher Scientific) plus 2 μg/mL propodium podide staining (Thermo Fisher Scientific), cells were rehydrated in 1% bovine serum albumin-PBS at a density of $1 \times 10^6$ cells/mL, transferred into flow cytometry analysis tubes, and stored overnight at 4 ℃ prior to samples analysis by Beckman Coulter FC500 Flow Cytometer. EdU signal was detected in FL1 (515–536 nm), and PI signal was detected in FL3 (590–615 nm). Twenty thousand events will be recorded and analyzed using the FlowJo software. In the Click-iT™ EdU Alexa Fluor™ 594 Imaging Assay, cells were mounted in ProLong® Gold Antifade Mountant for 24 h before imaging using the Leica fluorescence microscope in HCRI Tissue Core Facility (Notre Dame, IN, USA). All experiments were repeated three times, and quantitative results were analyzed by one-way analysis of variance (ANOVA) or t test, where $P < 0.05$ was considered statistically significant.

**Immunoprecipitation and immunoblotting assays.** For immunoprecipitation, cells first were transfected with either empty or DEDD plasmid. Forty-eight hours later, cells were treated with 1 ng/mL of EGF and then with 1 μM MG-132

for 6 h before collection. Harvested cells were re-suspended in 500 μL of M2 lysis buffer (50 mM Tris-HCl, pH 7.4, 150 mM NaCl, 1% Triton X-100, 0.5 mM EDTA). After incubation for 1 h on ice, cell lysates were incubated with 1 μg of Rb (4H1) antibody overnight (18 h). One percent of cell lysates were saved as input control. After that, the samples were incubated with 50 μL of protein A/G agarose beads (Santa Cruz) for 2 h at 4 ℃. Immunocomplexes were washed with M2 lysis buffer three times before immunoblotting. For immunoblotting, proteins were extracted using M2 lysis buffer and proteinase plus phosphatase inhibitor cocktail (Cell Signaling #5872). Protein extracts were resolved using 4–20% gradient SDS-polyacrylamide gel electrophoresis gels and transferred to nitrocellulose membranes (Thermo Fisher Scientific #88018). Membranes were probed with primary antibodies overnight on a 4 ℃ shaker and then incubated with horseradish peroxidase (HRP)-conjugated secondary antibodies. Signals were visualized with SuperSignal West Pico Chemiluminescent Substrate (Thermo Fisher Scientific #34078).

**Antibodies.** DEDD (Sc-271192) and HSC70 (sc-7298) antibodies were purchased from Santa Cruz Biotechnology. β-Actin (ab8226) was purchased from Abcam. FLAG M2 (F1804) was purchased from Sigma-Aldrich. Cyclin B1 (#4138), cyclin D1 (#2978), cyclin A2 (#4656), cyclin E1 (#4129), p-Erk (#4370), Erk (#4695), p-EGFR (#3777), EGFR (#4267), p-Akt (#4060), Akt (#4691), Rb (#9309), p-Rb (#9308), RBL1 (#89798), Ubi (#3933), p-FoxM1 (#14655), and FoxM1 (#5436) were purchased from Cell Signaling Technology. For immunoprecipitation assay, antibodies were diluted into 50 ng/μL. For immunoblotting assay, antibodies were diluted into 2.5 ng/μL. For immunofluorescence assay, antibodies were diluted into 25 ng/μL. All immunoblotting original uncropped images were provided as Supplementary materials (Source Data (pptx)) with the manuscript.

**Immunofluorescence assay.** For immunofluorescence staining, cells were serum starved overnight before stimulation with 1 ng/mL of EGF and treated with 1 μM MG-132 for 6 h before fixation. Cells were fixed with 4% paraformaldehyde (PFA) and permeabilization in 0.2% Triton X-100 (PBS) and followed by anti-FLAG and Rb antibodies (dilution 1:50) incubation for 18 h at 4 ℃. After incubation with tetramethylrhodamine-conjugated or fluorescein isothiocyanate-conjugated secondary antibody (dilution 1:100, 25 ng/μL) for 1 h at 25 ℃, images were captured with an Olympus FV1000 2-photon confocal microscope.

**Antibody-coupled Dynabeads immunoprecipitation.** Cells were infected with lentiviruses containing pLenti-CMV-puro-3×-FLAG-ΔNLS-DEDD plasmids for 72 h before puromycin (1 μg/mL) treatment for 48 h. The cells were further cultured for two more doublings. The harvested cells were washed with 1× PBS and re-suspended in 1:9 ratio of cell mass to extraction buffer (Dynabeads® Co-Immunoprecipitation (Co-IP) Kit, Thermo Scientific) with protease inhibitors. The cell lysates were incubated on ice for 30 min and then centrifuged at $2600 \times g$ for 5 min at 4 ℃ to remove large cell debris and nuclei. BCA (bicinchoninic acid) assay was used to quantify the protein concentration in the supernatant. About 5000 μg of protein was incubated with 1 μg of anti-FLAG antibody (Sigma-Aldrich, F1804)-coupled Dynabeads (10 μg of antibody) on a roller at 4 ℃ for 1 h. Another 2000 μg of protein was incubated with 1 mg of anti-mouse IgG1 (Cell Signaling, 5415S)-coupled Dynabeads (10 μg of antibody) for non-specific control immunoprecipitation. One percent of cell lysates were saved as input control. The Dynabeads® Co-IP complexes were washed with extraction buffer for three times. The proteins bound to beads were released in 60 μL of elution buffer provided by the kit. The proteins were added five volumes of cold (−20 ℃) acetone overnight at −20 ℃.

**TNBC in vivo xenograft experiments.** All animal experiments were performed in strict accordance with IACUC guidelines. Rag1 knockout (Rag1−/−) and non-obese diabetic/severe combined immunodeficiency (NOD SCID) mice were ordered from Taconic Farms (Hudson, NY, USA). For TNBC xenograft experiments, 8-week-old female Rag1−/− mice were transplanted with one million cancer cells suspended in 0.1 ml of Matrigel mixture (1:1 mixture ratio of Matrigel and DMEM/F-12 medium without FBS) orthotopically into contralateral sides of the fourth mammary fat pads of the mice. For TNBC PDXs, 3.5-week-old female NOD SCID mice have received PDX tumor pieces as previously described[41]. When the tumor reached the volume of approximately 100 mm³, the mice were randomly divided into groups with an even distribution of tumor sizes (10 mice/group). LAP was given every 48 h at a dose of 100 mg/kg in the vehicle (0.5% hydroxypropylmethylcellulose with 0.1% Tween-20) via oral gavage. Abemaciclib was given every 48 h at a dose of 25 mg/kg in the vehicle (0.5% hydroxypropylmethylcellulose with 0.1% Tween-20) via oral gavage. Tumor volume was calculated as: tumor volume $= 1/2$ (length × width²). As designated time point post treatment, tumors were collected and fixed in 4% PFA for 24 h followed by 70% ethanol for another 24 h before formalin-fixed paraffin-embedded embedding for histology analysis.

**Immunohistochemical analysis.** PFA-fixed tumor samples were paraffin embedded and cut into 5 μm thin slices and fixed on glass slides. DEDD, Ki-67, or cleaved caspase-3 were detected by DEDD antibody (Santa Cruz Biotechnology

#sc-271192), Ki-67 antibody (Ki-67 (D2H10), rabbit monoclonal antibody (IHC-specific) (Cell Signaling Technology #9027), or cleaved caspase-3 antibody (cleaved caspase-3 (Asp175) antibody, Cell Signaling Technology #9661) through IHC using the Polink-2 Plus HRP Rabbit with DAB Kit (GBI Labs) after 20 min of antigen retrieval. The stained slides were scanned using the Aperio scanner in the HCRI Tissue Core Facility (Notre Dame, IN, USA). Images were scored for the percentage of Ki-67 tumor cell nuclei per total tumor cell nuclei in each captured field using the ImageScope software. For *DEDD* and cleaved caspase-3, images were scored as the percentage of positive staining area per captured total area. All quantification was performed in a fashion that was blinded to the treatment group.

**Drug synergism analysis**. The raw data for drug synergism analysis were generated by the MTT assay. The cells were evenly plated 24 h before administration of drug dosages. Both single and combination treatments were completed in triplicate at each dosage (4–6 different dosages in a series dilution). Then, the absorbance reading of each well was calculated as FA by the treatment. The wells treated with DMSO served as non-affected controls. The wells with medium only served as full-affected control. The mean FAs at different concentrations were then analyzed by the CompuSyn software to generate simulation for the drug synergism analysis. The CI were plotted along with indicated FA for each cell line. CI > 1, antagonism; CI = 1, addition; CI < 1, synergism.

**Bioinformatics and statistical analysis**. TCGA breast-invasive carcinoma provisional dataset (1105 samples) was sourced from cBioportal [http://www.cbioportal.org]. The oncoprint heatmap was generated by including mutations, putative copy-number changes, and mRNA expression z-scores (RNA-seq V2 RSEM with $z$ threshold ± 2). The survival data were extracted from the same dataset for Kaplan–Meier plot in R using "Survival" and "Survminer" packages. The boxplot of the TCGA segmented copy-number changes without germline CNV ($n = 1099$) was generated using the University of California, Santa Cruz (UCSC) Cancer Browser. Gene expression data of commonly used cells lines were extracted from Broad Institute Cancer Cell Line Encyclopedia (CCLE). The heatmap of relative gene expression was generated in R using pHeatmap package. RNA-seq profiling of *DEDD* knockdown cells was conducted by standard Illumina TruSeq library preparation followed by Illumina sequencing using miSeq. The differential gene expression analysis was performed using DE-seq2 package. An adjusted $P$ value of <0.05 was considered statistically significant. We performed GSEA analysis using GenePattern interface [http://www.broadinstitute.org/gsea/] using C6 v5.2 symbol (oncogenic signature) sub-datasets. Network analysis was conducted through the web-based bioinformatics package NetworkAnalyst [https://www.networkanalyst.ca/]-based on recommended protocol. All quantitative data were analyzed and plotted using the GraphPad Prism software. $P$ values were generated using either two-sided $t$ test or Fisher's exact test, or one-way ANOVA test.

**Reporting summary**. Further information on research design is available in the Nature Research Reporting Summary linked to this article.

## Data availability
All sequencing data that support the findings of this study have been deposited in the National Center for Biotechnology Information Gene Expression Omnibus (GEO) and are accessible through the GEO Series accession number GSE131303. All other relevant data are available from the corresponding author on request.

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

## Acknowledgements

This work was supported by the Department of Defense through the Breast Cancer Research Program under Award W81XWH-18-1-0473. Opinions, interpretations, conclusions, and recommendations are those of the author and are not necessarily endorsed by the Department of Defense. This work was partially supported by NIH R01 CA194697-01 (S.Z.), NIH R01 CA222405-01A1 (S.Z.), and NIH CTSI core facility pilot grants (S.Z.). We would additionally like to acknowledge and thank the Dee Family endowment (S.Z.). We would also like to thank Zhang Lab members for scientific insight and support. For insightful technical assistance, we thank Charles R. Tessier, Ph.D. We would also like to thank Dr. Nakshatri (IUSM) for offering precious KTB cell lines and intellectual discussions of the manuscript. For PDX samples, we would like to thank Dr. Littlepage (ND) for offering tumor samples and Dr. Welm (Utah) for tremendous efforts in establishing PDX model. We are additionally grateful for the use of the following core facilities: Notre Dame Genomics and Bioinformatics Core Facility, Notre Dame Freimann Life Sciences Center, Harper Cancer Research Institute Biorepository, Indiana University School of Medicine South Bend Imaging and Flow Cytometry Core, KTB, and Indiana University Simon Cancer Center Core Facility.

## Author contributions

Y.N. and S.Z. conceived the original hypothesis and designed experiments. Y.N., K.R.S., B.A.W., J.K.K., I.H.G., P.M.S., X.T., L.J., M.H., L.S., J.W., L.E.L., H.N., and S.Z. performed experiments. Y.N., K.R.S., B.A.W., J.K.K., P.M.S., E.N.H., and S.Z. analyzed data. Y.N., I.H.G., L.E.L., H.N., and S.Z. wrote and revised the manuscript. S.Z. supervised the study.

## Additional information

**Competing interests:** The authors declare no competing interests.

