## [Peer Review File · Nature Communications]

Reviewers' comments:

Reviewer #1 (Remarks to the Author):

In this manuscript, "Functional Screen Identifies Triple-Negative Breast Cancer Vulnerability Driven by Death Effector Domain-Containing Protein" by Ni Y et al., a barcoded RNAi screening was utilized to discover drug-resistant tumor-promoting genes in HER2-/ER-/PR- breast cancer cell lines. The authors focused on one gene, DEDD, by detecting the most prominent (67%) genetic alteration in the database (The Cancer Genome Atlas) of triple-negative breast tumors among 10 genes hits in the RNAi screening. They used 8 different breast cancer cell lines and 4 normal breast epithelial cell lines in knocking down or overexpressing DEDD to show a novel role for DEDD in cell cycle progression. First, they showed the pulsed increase in DEDD expression along with cyclin D1 during G1-S transition, as well as the association of DEDD protein with HSC70 and Rb, the latter of which leads to the degradation of Rb protein and following cell cycle progression. Second, combinations of lapatinib and different G1 CDK inhibitors were tested for the synergetic tumor-suppressing effects on the breast cancer cell lines with/without silencing or overexpressing DEDD. Lastly, mice bearing the xenografts of the breast cancer cell lines were treated with lapatinib in combination with the CDK4/6 inhibitor abemaciclib, showing the synergetic tumor-suppressing effects in vivo as similarly seen in vitro. From this study, the authors inferred a possibility of the presumably ignored efficacy of applying CDK4/6 inhibitor for triple-negative breast cancers lacking Rb expression.

One newly found molecular function of DEDD for cell cycle regulation in this study is that DEDD accelerates G1-S progression by indirect stabilization of cyclin D1 via association with HSC70, in contrast to a previous report that showed a suppressive role for DEDD in G2-M transition via direct interaction with cyclin B1 and Cdk1 (Arai S et al., PNAS 2007 (Ref.23 in the manuscript)). This may indicate an intrinsic role for DEDD in the triple-negative breast cancer cells, considering the remarkable cytosolic, but not nuclear, localization (Fig. 3A in the manuscript). Alternatively, this finding may link a missing mechanism of cyclin D3/cdk4/cdk6 stabilization by DEDD in uterine endometrial cells (Mori M et al., JCI 2011).

Although the presented experimental data and meta-analysis of database resources are appreciately in part, there are many issues to be further addressed to meet the quality for the publication in Nat Commun.

1. In Fig. 2E, the protein level of DEDD seems to be correlated with that of cyclin D1 but not the others. However, the annotation for the cell cycle from M phase to S phase depends only on the histograms of flow cytometry (Fig. 2F, PI staining?), which does not distinguish G2/M phases.
2. The western blot in Fig.2G is not clear enough to show the corresponding reduction of cyclin D1 by knocking down DEDD. Instead of MDA-MB-468, it might be better to use HCC1806 or HCC38, which show more clear bands in Fig. 2E.
3. Similarly, Fig. 3J is important to show the reduction of cyclin D1 by knocking down HSC70, but the results are not clear in either KTB-34 or KTB-37 cells. Probably this experiment lacks the synchronization, as the blot in Fig.3G is more prominent.
4. In Fig.5D, it is not acceptable to use an arbitrary score for the Combination Index (CI), which should be between 0 and 1, when the drugs have synergistic effects. There is no description how this arbitrary unit was determined or calculated. A particular dose for each drug should be used to compare different cell lines. The dose and the name of the drugs should also be annotated.
5. Fig.6C-J and Supplementary Fig.8 show in vivo data for a strong synergistic effect of lapatinib and abemaciclib on HCC1806 xenografts and milder synergistic effects on MDA-MB-468 and HCl-001 xenografts. However, while MDA-MB-468 showed higher DEDD expression compared to HCC1806 (Fig.2A), the xenografts showed mild and similar DEDD expression between these two cell lines (Fig. 6C). This is a weakness of this manuscript so as to insist the relevance of DEDD-driven vulnerability to a single drug and the DEDD-dependent synergistic effects of the drugs. Also the method for DEDD IHC is missing.

Minor concerns are shown below.

6. There are a lot of grammatical problems throughout the manuscript. Consulting with an English reviewer is highly recommended.
7. In Materials and Method, the resource information for human DEDD cDNA is missing.
8. Fig. 1F is missing in the text: "shDEDD showed the most consistent and significant effect of sensitizing TNBC cells to the Lapatinib treatment..."
9. Supplementary Fig. 1C-D shows the diminished Akt and ERK phosphorylation by lapatinib treatment. However, precisely saying, it is necessary to show both time points (2 and 24 hrs) on one membrane, when the authors want to show the re-activation of Akt.
10. It is strange to say that the "cytosol DEDD expression promoted cell proliferation in HMECs under the stress of cell cycle disruption," since Nocodazole-treated cells are arrested at G2-M phase and cannot divide. Supplementary Fig. 5D may have resulted from the initially increased cell proliferation with DEDD overexpression before or at the time of adding Nocodazole.
11. Fig. 6C shows DEDD staining in the xenografts. However, since the amino acid sequences for murine and human DEDD are highly identical (>98%), a cross reactivity of the antibody to any infiltrating murine cell is concerned. This can be solved either by co-staining with human/mouse-specific marker(s) or by using a negative control tissue without xenograft.

Reviewer #2 (Remarks to the Author):

Developing novel therapeutic strategies for patients with TNBC is a crucial question that drives many streams of research. In this manuscript, Ni et al have elegantly shown that gain-of-DEDD drives a cancer-specific vulnerability to CDK4/6 inhibition that is usually used for Rb-high breast cancers. While searching for potential synthetic lethality interactions with EGFR/HER2 signalling, the authors uncovered three genes depleted under Lapatinib treatments. Among them they validated the effects of DEDD in vitro and showed that DEDD knockdown led to inhibition of DEDD-high TNBC cells. They then demonstrated that cytosolic DEDD facilitates G1/S transition which is mediated by binding to HSC70 and promotes proteasome degradation of Rb proteins.

Since Rb expression is a prerequisite for CDK4/6 inhibitor use in the clinic, most TNBC tumours are not treated with these inhibitors as Rb status is low. The authors showed that independent of Rb status, DEDD-high TNBC cells were vulnerable to CDK4/6 inhibitors in combination with Lapatinib. This study shows for the first time, the potential clinical utility of CDK4/6 and EGFR/HER2 inhibition in DEDD-high TNBC tumours, hence providing a novel avenue for therapy.

I am mindful that the paper has a huge amount of data but there are minor concerns that should be addressed (in my opinion) before publication.

1. The authors consistently state in their figure legends that P values were analysed by t-test (two-tailed). In most instances, they have more than three groups in their dataset. This increases the chance of a Type I error if t-test are used and hence the more appropriate test is a one-way ANOVA. Can the authors please re-test all their significance values to ensure that the data is still significant?
2. The authors state that multiple shRNAs were used to target DEDD, ABCB10 or UAP1 in three TNBC cell lines. This is only the case for DEDD and not done for the other genes. Can the authors explain why? or Show the data in supplementary figures.
3. Although the results of ABCB10 and UAP1 were moderate, their deletion still sensitises TNBC cells to Lapatinib treatment. Can the authors provide an explanation of how these genes function

and why they chose to focus on DEDD? Also a section in the discussion on this point will be useful.

4. The accumulation of RBL1 (p107) was only noticed in shDEDD-1. However in shDEDD-2 there is a clear increase of RBL1 (p107). Can the authors explain this and have they tried any other shRNAs?

Minor

1. Supplementary Fig 1. A – What do the red highlighted cell lines refer to?
2. Supplementary figure 3 and Fig 1e not referenced adequately in text.
3. Result section: HSC70 mediated DEDD-dependent cycD1 exp and G1/S expression. Line 18, 'NLS-DEDD overexpressed cells' missing reference of Figure 3J.
4. Fig 5F. How long were the cells treated?
5. Discussion: Chromosome 1q gene DEDD facilitates TNBC progression. Line 12. Presence instead of present.

Reviewer #3 (Remarks to the Author):

The manuscript by Ni et al. describes the identification of DEDD as a synthetic lethal interaction with lapatinib in TNBC through the use of a shRNA screen in HCC1806. Hits from the screen were prioritized based on increased expression in TNBC. They show that DEDD (over)expressing TNBC cells are sensitive to DEDD depletion. As potential mechanism of action they describe that cytosolic DEDD upregulates cyclin D1 expression through a potential interaction with HSC70. In addition, they propose that DEDD also interacts with RB and RBL1 to promote their degradation. These results led to the hypothesis that the combination of CDK4/6 and EGFR inhibition should be effective in DEDD overexpressing TNBC independent of RB status. They demonstrate the effectiveness in TNBC cell line proliferation and tumor growth providing rational for a potential clinical application of this combination.

With respect to novelty, the combination of CDK4/6 inhibitors (abemaciclib, palbociclib) with EGFR (cetuximab), HER2 (lapatinib) or MEK (PD-0332991) inhibitors has been described before (for example in REF 53) and are currently studied in clinical trials in different tumor types. The use of DEDD expression as potential biomarker for the combination of Abemaciclib and Lapatinib in TNBC is questionable as cell lines with low expression are also sensitive to the combination (HCC1806, KBT-21, KBT-37; figure 5E,6B). This leaves doubt about the dependency on DEDD overexpression and the potential therapeutic window for this combination. In addition, the work does not provide an mechanistic explanation for the observed synergy between the two treatments, why does abemaciclib alone increase pERK and does it reduce pAKT in the presence of lapatinib (figure 5C)?.

The authors have studied the effects of knock-down, over-expression and inhibitor treatment in many different cell lines and models. However, in many instances, observations are first made in one cell line and subsequently further explored in other cell lines. This makes it difficult to interpret the data and draw strong conclusions. This is exemplified by Figure 3H, 3J and K, and the absence of the PDX HCI-004. This inconsistency is also observed in different figures testing the same perturbation or condition, for example, the lack of an increase in RB levels by knockdown of DEDD in figure 4C compared to figure 4A. Also, why is the overexpression of FLAG-dNLS-DEDD not downregulating RB expression (in the 1% input lane) in figure 4F compared to figure 4D.

A major concern with this work is that most of the observations are based on either knock-down using shRNA or over-expression of DEDD or DEDD mutants. Especially there is concern about the reproducibility of the effects of shRNA knock-down with other means of perturbation. The authors should take caution in using high titer lentiviruses encoding shRNAs and preferably perform the experiments at low MOI to ensure single integrations. In addition, the authors should validate the results with CRISPR-based inactivation (e.g. CRISPRi or a conditional intron and CRE recombinase). It is disconcerting that the large-scale efforts across many cell lines for dependencies do not show an effect of loss of DEDD expression. In addition there seems to be little correlation with knock-down and the effect-size. For example, in Figure S3a, there seems a lack of correlation with knock-down, shABCB10-5 best enhancer with the least knock-down? In Figure S3b, there is a very small effect of complete knock-down and in Figure S3c and Figure 2F, only 2 shRNAs were tested for effect, whereas for all others 5 shRNAs were used? Also in Figure S3C, protein expression should be provided in addition to mRNA expression. This in relation to Figure 4A and B describing dependency upon knock-down.

The basis of this work is an shRNA screen but the details of the screen and the interpretation of the screen is insufficient and not presented in a proper way. The list is an alphabetical sorted list of genes. The authors should at least provide raw and processed data of the individual replicates, individual shRNA depletion scores, and some sort of method to score the behavior of shRNAs targeting the same gene (e.g. RRA or MAGeCK). Finally, this list should provide fold depletion in treated and untreated conditions accompanied by statistical significance.

Other remarks:

Figure 1 f, normalized cell to number to what, DMSO control for each shRNA? Cannot exclude strong effect on untreated cells (as for MDA-MB-468 in figure 2C).

What means the annotation of the axis in figure 3E; 100%?

Supplemental Figure 4 B what is the FDR cut-off?

How were the correlations to DEDD levels in figure 2E determined? Quantitative? Normalized to beta-Actin (seems overexposed and not quantitative?)

Evidence for cytoplasmic localization insufficient. The authors should show the analysis of TNBC cells lines with different levels of DEDD for the localization by both IHC and cell fractionation followed by western blotting.

The effects of overexpression of DEDD or dNLS-DEDD are the same on cell cycle progression. How can the authors conclude that cytoplasmic localization of DEDD is responsible for the observed phenotype?

In Figure 3J levels of DEDD are strongly reduced by siHSC70 and this could readily explain the reduction in S-phase increase in Figure 3i.

The role of HSC70 in the observed phenotypes can be the consequence of misfolded over-expressed proteins that are bound and stabilized by HSC70.

The authors should provide the expression levels of dNLS-DEDD in KBT-21 for figure 5E as this can be very low as can be seen in figure 4E, no MG-1320.

What about Xenograft HCI-004? The authors should provide growth curves of both PDX models in absence and presence of treatment similar to figure 6E.

Point-to-Point Answers to Reviewers' Critiques

Re: Manuscript # NCOMMS-18-32121

We would like to thank the editor and the reviewers for valuable and constructive comments. In this revised manuscript, we have carefully addressed each of the reviewers' comments in full by adding a significant amount of new data (24 new panels), additional descriptions of methods and a new section of Discussion. Here, we have indicated the changes made to the manuscript to take into account the comments from the three reviewers, which are reproduced below in *italics (font size 10)* for your easy reference. Our point-by-point replies to every comment from the reviewers are as follows:

Reviewer #1 (Remarks to the Author):

Overall summary:

... One newly found molecular function of DEDD for cell cycle regulation in this study is that DEDD accelerates G1-S progression by indirect stabilization of cyclin D1 via association with HSC70, in contrast to a previous report that showed a suppressive role for DEDD in G2-M transition via direct interaction with cyclin B1 and Cdk1 (Arai S et al., PNAS 2007 (Ref.23 in the manuscript)). This may indicate an intrinsic role for DEDD in the triple-negative breast cancer cells, considering the remarkable cytosolic, but not nuclear, localization (Fig. 3A in the manuscript). Alternatively, this finding may link a missing mechanism of cyclin D3/cdk4/cdk6 stabilization by DEDD in uterine endometrial cells (Mori M et al., JCI 2011).

Answer:

We sincerely appreciate the reviewer's insights in putting our new findings along the previous literature. Indeed, our study provided additional mechanistic explanations to multifaceted cell cycle regulatory function of DEDD. We have added the following discussion and added relevant literature in this revised manuscript discussion section "Multifaceted mechanisms of DEDD-mediated G1/S transition." (See manuscript main text, page 18, line 20 - page 19, line 2).

Major comments:

1. In Fig. 2E, the protein level of DEDD seems to be correlated with that of cyclin D1 but not the others. However, the annotation for the cell cycle from M phase to S phase depends only on the histograms of flow cytometry (Fig. 2F, PI staining?), which does not distinguish G2/M phases.

Answer:

We thank the reviewer for this comment. To more precisely annotate cell cycle stages for this experiment, in this revised manuscript, we employed EdU incorporation assay to label S-phase (HCC1806 cells and MDA-MB-468 cells, new Supplementary Fig. 5a). After the EdU and PI staining, cells were sorted by FACS (new Supplementary Fig. 5a) into sub-populations as G0/G1 (P1, EdU low and PI low), S (P2, EdU high and PI moderate) and G2/M (EdU low and PI high). Then the sorted three cell populations of each TNBC cell line were IF stained with anti-DEDD antibody and DEDD positive cells were counted from each cell cycle phase. We observed DEDD positive cells are significantly higher (around 20%) in G0/G1 phase, but not in S or G2/M phase in HCC1806 cells. In MDA-MB-468 cells, DEDD positive cells are high in G0/G1 phase (around 35%), moderate in G2/M phase (around 15%) and low in S phase (around 3%). This result indicated that DEDD primarily expressed in G1 phase. In addition, we also quantified DEDD expression levels with different cyclin expression levels at different time points in multiple TNBC cell lines (new Supplementary Figure 5d). In HCC1806 and HCC38 cells, the expression dynamic pattern of DEDD is similar to cyclin D1, but not other cyclins, suggesting a correlative expression between DEDD and cyclin D1. Finally, to examine the causal role of DEDD expression in the regulation of different cyclins, we performed siRNA DEDD knockdown experiments in both HCC38 and HCC1806 cells in addition to MDA-MB-468. As shown in the new Fig. 2f, knocking down DEDD strongly reduced cyclin D1 level and slightly reduced cyclin B1 expression.

2. The western blot in Fig.2G is not clear enough to show the corresponding reduction of cyclin D1 by knocking down DEDD. Instead of MDA-MB-468, it might be better to use HCC1806 or HCC38, which show more clear bands in Fig. 2E.

Answer:

We thank the reviewer for this valuable suggestion. As a reviewer suggested, we generated new data in both HCC38 and HCC1806 cells (new Fig. 2f). We observed that, consistent with MDA-MB-468 results, siDEDD efficiently decrease DEDD expression in TNBC cell lines and resulted in a significant decrease of cyclin D1 expression in both serum-starved condition (Fig. 2f, 0 hr) and after FBS stimulation (Fig. 2f, 9 hrs in HCC38 and 6 hrs in HCC1806, 6 hrs in MDA-MB-468). These consistent results indicate that DEDD is an upstream regulator of cyclin D1.

3. Similarly, Fig. 3J is important to show the reduction of cyclin D1 by knocking down HSC70, but the results are not clear in either KTB-34 or KTB-37 cells. Probably this experiment lacks the synchronization, as the blot in Fig.3G is more prominent.

Answer:

In this revised manuscript, we have reprobated cyclin D1 changes upon Δ NLS-DEDD expression and siHSC70 treatment in both KTB-34 and KTB-37 cells (new Fig. 3i, cyclin D1). The updated Fig. 3i shows a much clearer evidence that overexpression of Δ NLS-DEDD in KTB cells significantly enhances the cyclin D1 expression, and this enhancement is depending on HSC70, evidenced by which siHSC70 treatment significantly reduces cyclin D1 expression caused by Δ NLS-DEDD overexpression (Fig. 3i, j).

4. In Fig.5D, it is not acceptable to use an arbitrary score for the Combination Index (CI), which should be between 0 and 1, when the drugs have synergistic effects. There is no description how this arbitrary unit was determined or calculated. A particular dose for each drug should be used to compare different cell lines. The dose and the name of the drugs should also be annotated.

Answer:

We agree with the reviewer on this comment. In this revised manuscript, we treated TNBC cells under the same drug dosage (new Fig. 5a, lapatinib: 1 μ M; abemaciclib: 0.5 μ M). In the new Fig. 5a, we summarized this new experiment by showing cell viability changes of multiple TNBC cell lines with different DEDD expression levels in response to designated treatments. By cross-comparing to DEDD expression level in TNBC cell line (new Fig. 2a), we observed a trend of positive correlation between DEDD expression level and TNBC cell's sensitivity to combination treatment.

5. Fig.6C-J and Supplementary Fig.8 show *in vivo* data for a strong synergistic effect of lapatinib and abemaciclib on HCC1806 xenografts and milder synergistic effects on MDA-MB-468 and HCl-001 xenografts. However, while MDA-MB-468 showed higher DEDD expression compared to HCC1806 (Fig.2A), the xenografts showed mild and similar DEDD expression between these two cell lines (Fig. 6C). This is a weakness of this manuscript so as to insist the relevance of DEDD-driven vulnerability to a single drug and the DEDD-dependent synergistic effects of the drugs. Also the method for DEDD IHC is missing.

Answer:

First, based on our new data generated in Fig. 5a, TNBC cell lines with a higher DEDD expression (HCC38, MDA-MB-468 and BT-20) appears to be more sensitive to the combination treatment, although all tested TNBC cell lines except MDA-MB-231 respond to the combination treatment (combo). Such correlation between DEDD expression and sensitivity to combo treatment suggested a dependency of DEDD level of combo sensitivity.

Second, beyond above correlative observation, to further demonstrated the functional dependency of DEDD expression pertains to the sensitivity to the CDK4/6 inhibitor-containing regimen, we modulated the DEDD expression in both normal mammary tissue cell lines (KTB) as well as TNBC cells (MDA-MB-436) with a lower expression of DEDD. As shown in Fig. 5d-g, ectopic expression of the cytosol DEDD in KTB cells or overexpression of cytosol DEDD in MDA-MB-436 cells rendered above cell more sensitive to both Abemaciclib treatment alone and combo treatment. Such increased sensitivity to

combo, as a direct result of DEDD overexpression, indicated a causal role of DEDD level in shaping the overall sensitivity of a given cell to combo treatment in vitro.

Considering the in vivo drug response could be a result of multiple factors, we agree with reviewer's comments and have tuned down our conclusion regarding the DEDD's predictive power in drug response in vivo and clinical implications as a biomarker in the updated "Discussion".

Lastly, we have added the DEDD IHC method in this revised manuscript "Materials and Methods" section (manuscript main text, page 29, line 17-19).

Minor concerns are shown below.

6. *There are a lot of grammatical problems throughout the manuscript. Consulting with an English reviewer is highly recommended.*

Answer:

We thank the reviewer's comment. We have thoroughly proof-read this revised manuscript.

7. *In Materials and Method, the resource information for human DEDD cDNA is missing.*

Answer:

Human DEDD cDNA information has been added into the updated manuscript "Materials and Methods" section, right after cell culture paragraph, Page 23, line 5-Page 24, line 2.

8. *Fig. 1F is missing in the text: "shDEDD showed the most consistent and significant effect of sensitizing TNBC cells to the Lapatinib treatment..."*

Answer:

The description of Figure 1F has been added to the updated manuscript text (manuscript main text, page 6, line 16).

9. *Supplementary Fig. 1C-D shows the diminished Akt and ERK phosphorylation by lapatinib treatment. However, precisely saying, it is necessary to show both time points (2 and 24 hrs) on one membrane, when the authors want to show the re-activation of Akt.*

Answer:

We agree with the reviewer for this comment. We have revised our terminology when describing this result in our revised manuscript. To be more precise, the Supplementary Fig. 1c demonstrated that lapatinib not as effective in inhibiting p-AKT at 24 hrs compared to 2 hrs. In addition, we provided additional data in this revised manuscript showing knocking down DEDD enhanced lapatinib's activity in suppressing p-AKT at 24 hrs (new Supplementary Fig. 3f.)

10. *It is strange to say that the "cytosol DEDD expression promoted cell proliferation in HMECs under the stress of cell cycle disruption," since Nocodazole-treated cells are arrested at G2-M phase and cannot divide. Supplementary Fig. 5D may have resulted from the initially increased cell proliferation with DEDD overexpression before or at the time of adding Nocodazole.*

Answer:

In this revised manuscript, we have completely rewritten the section to describe the phenotype. Please see revised manuscript text page 9, line 13-15.

11. *Fig.6C shows DEDD staining in the xenografts. However, since the amino acid sequences for murine and human DEDD are highly identical (>98%), a cross-reactivity of the antibody to any infiltrating murine cell is concerned. This can be solved either by co-staining with the human/mouse-specific marker(s) or by using a negative control tissue without xenograft.*

Answer:

To examine the potential antibody cross-reaction, we have performed IHC staining of DEDD in negative control tissue without TNBC xenograft as the reviewer suggested. We did observe very light staining of

mouse mammary gland epithelial cells. However, the staining intensity is significantly weaker than tumor cells. Besides, the human tumor cells have a significantly bigger nucleus. When we semi-quantitatively evaluated the IHC staining, we only focused on xenografted tumor cells. The new IHC control tissue data have been provided side by side with DEDD staining of xenografts tissues in the new Fig. 6b.

Reviewer #2 (Remarks to the Author):

Overall summary:

... Since Rb expression is a prerequisite for CDK4/6 inhibitor use in the clinic, most TNBC tumours are not treated with these inhibitors as Rb status is low. The authors showed that independent of Rb status, DEDD-high TNBC cells were vulnerable to CDK4/6 inhibitors in combination with Lapatinib. This study shows for the first time, the potential clinical utility of CDK4/6 and EGFR/HER2 inhibition in DEDD-high TNBC tumours, hence providing a novel avenue for therapy.

Answer:

We thank the reviewer for appreciating the potential translational impact of our study.

1. The authors consistently state in their Figure legends that P values were analyzed by t-test (two-tailed). In most instances, they have more than three groups in their dataset. This increases the chance of a Type I error if t-test are used and hence the more appropriate test is a one-way ANOVA. Can the authors please re-test all their significance values to ensure that the data is still significant?

Answer:

Experiments in Fig. 2 were originally tested for One-way ANOVA. Other experiments were re-tested for One-way ANOVA Dunnett's Multiple Comparison Test. In this revised manuscript, all Figures have been updated with the new p values. Here, we listed representative test results for reviewer's easy reference.

MTT Cell Viability Test (Fig. 1F):

HCC1806

Dunnett's Multiple Comparison Test	Mean Diff.	q	Significant? P < 0.05?	Summary	95% CI of diff
plko.1 vs shDEDD-1	0.3208	3.341	Yes	*	0.02322 to 0.6184
plko.1 vs shDEDD-2	0.3022	3.148	Yes	*	0.004650 to 0.5998
plko.1 vs shDEDD-3	0.3114	3.244	Yes	*	0.01384 to 0.6090

HCC38

Dunnett's Multiple Comparison Test	Mean Diff.	q	Significant? P < 0.05?	Summary	95% CI of diff
plko.1 vs shDEDD-1	0.5389	5.579	Yes	**	0.2395 to 0.8383
plko.1 vs shDEDD-2	0.7480	7.745	Yes	***	0.4487 to 1.047
plko.1 vs shDEDD-3	0.6623	6.858	Yes	**	0.3630 to 0.9617

MDA-MB-468

Dunnett's Multiple Comparison Test	Mean Diff.	q	Significant? P < 0.05?	Summary	95% CI of diff
plko.1 vs shDEDD-1	0.4653	4.269	Yes	*	0.1275 to 0.8031
plko.1 vs shDEDD-2	0.4244	3.894	Yes	*	0.08656 to 0.7622
plko.1 vs shDEDD-3	0.5117	4.695	Yes	**	0.1739 to 0.8495

Crystal Violet Assay (Fig. 6A):

HCC38

Dunnett's Multiple Comparison Test	Mean Diff.	q	Significant? P < 0.05?	Summary	95% CI of diff
DMSO vs LAP	-36.47	2.697	No	ns	-75.41 to 2.467
DMSO vs ABE	0.1013	0.007489	No	ns	-38.84 to 39.04
DMSO vs COMBO	-90.96	6.727	Yes	***	-129.9 to -52.02

MDA-MB-468

Dunnett's Multiple Comparison Test	Mean Diff.	q	Significant? P < 0.05?	Summary	95% CI of diff
DMSO vs LAP	-60.37	5.159	Yes	**	-94.06 to -26.67
DMSO vs ABE	-33.60	2.872	No	ns	-67.29 to 0.09317
DMSO vs COMBO	-89.04	7.610	Yes	***	-122.7 to -55.34

HCC1806

Dunnett's Multiple Comparison Test	Mean Diff.	q	Significant? P < 0.05?	Summary	95% CI of diff
DMSO vs LAP	11.28	2.638	Yes	*	0.4166 to 22.14
DMSO vs ABE	-47.51	11.11	Yes	***	-58.37 to -36.65
DMSO vs COMBO	-91.44	21.39	Yes	***	-102.3 to -80.57

2. The authors state that multiple shRNAs were used to target DEDD, ABCB10 or UAP1 in three TNBC cell lines. This is only the case for DEDD and not done for the other genes. Can the authors explain why? or Show the data in supplementary Figures.

Answer:

We are sorry for the confusion. In this current study, we only focused on studying DEDD's role in TNBCs. Our genomic screen was performed in HCC1806 cells. Since the biological variation of the effects of different shRNA in different cell lines is expected, we believe that one of the very important steps of shRNA screening experiment is to independently validate the screening hits before the further mechanistic study. When we conducted independent shRNA validation using multiple TNBC cell lines, we didn't observe a consistent pattern that knocking down of ABCB10 or UAP1 re-sensitize TNBC cell lines to the lapatinib treatment (Supplementary Figure 3b, c, d and e). Although all three shRNAs effectively knockdown ABCB10 or UAP1 in two TNBC cell lines, only two shRNAs can resensitize HCC1806 cells to the lapatinib treatment, while none of them show a significant effect on MDA-MB-468 cells (Supplementary Figure c, e). Compared to ABCB10 and UAP1, knockdown of DEDD showed consistent synergistic effective with all three TNBC cell lines (Fig. 1f). Therefore, we chose DEDD as our focus for this manuscript. We have rewritten our manuscript to clarify this point (See manuscript main text, page 6, line 11-20)

3. Although the results of ABCB10 and UAP1 were moderate, their deletion still sensitizes TNBC cells to Laptinib treatment. Can the authors provide an explanation of how these genes function and why they chose to focus on DEDD? Also a section in the discussion on this point will be useful.

Answer:

We are also curious about the function of ABCB10 and UAP1. In the independent validation experiments, knocking down either ABCB10 or UAP1 only led to a ~5-20% decrease of cell viability under lapatinib treatments, which are significantly less effective comparing to shDEDD (more than > 40% decrease). This may indicate that compare to DEDD, both ABCB10 and UAP1 have less contribution to the resistance to lapatinib treatment in HCC1806 cells. Considering the reasons that have been stated in our answers to critique #2 above, we decided to focus on DEDD only in this study.

4. The accumulation of RBL1 (p107) was only noticed in shDEDD-1. However, in shDEDD-2 there is a clear increase of RBL1 (p107). Can the authors explain this and have they tried any other shRNAs?

Answer:

In Fig. 4h, an increase RBL1 (p107) protein level has been seen in all three shDEDD groups. Under the MG-132 treatment (Fig. 4i), we observed a time-dependent accumulation of RBL1(p107) in pLKO.1 group. Under shDEDD, this trend of time-dependent accumulation of RBL1(p107) is significantly diminished, suggesting DEDD is responsible for RBL1(p107) protein turnover.

Minor Comments

1. Supplementary Fig 1. A – What do the red highlighted cell lines refer to?

Answer:

We have changed the colors of those cell lines. See in updated Supplementary Fig. 1a.

2. Supplementary Fig. 3 and Fig. 1e not referenced adequately in text.

Answer:

The manuscript has been updated to refer above to Figures. For Supplementary Fig. 3, Please see updated manuscript text page 6, line 11. For Fig. 1e, please see updated manuscript text page 6, line 3.

3. Result section: HSC70 mediated DEDD-dependent cycD1 exp and G1/S expression. Line 18, 'NLS-DEDD overexpressed cells' missing reference of Figure 3J.

Answer:

The manuscript has been updated. Please see updated manuscript text page 10, line 18.

4. Fig 5F. How long were the cells treated?

Answer:

Please see updated Fig. 5e, cells were treated for 24 hours.

5. Discussion: Chromosome 1q gene DEDD facilitates TNBC progression. Line 12. Presence instead of present.

Answer:

The manuscript has been updated. Please see updated manuscript page 16, line 18.

Reviewer #3 (Remarks to the Author):

Major comments:

1. With respect to novelty, the combination of CDK4/6 inhibitors (abemaciclib, palbociclib) with EGFR (cetuximab), HER2 (lapatinib) or MEK (PD-0332991) inhibitors has been described before (for example in REF 53) and are currently studied in clinical trials in different tumor types.

Answer:

We are aware of the previous literature describing the combinatorial therapies of CDK4/6 inhibitors and EGFR, HER2 or MEK inhibitors. However, while above combinatorial treatments with CDK4/6 inhibitor have been explored (e.g. previous manuscript REF 53), previous research has been performed in an Rb-wild type tissue context. In the case of breast cancer, all current clinical trials of CDK4/6 inhibitor (single or combination) are conducted in Rb-wild type ER+ or ER+/HER2+ breast cancer patients, NOT in triple-negative breast cancer (TNBC) (as noted by reviewer 2).

Rb-loss is frequently seen in TNBCs patients^{1,2}. Due to the prevailing presumption that CDK4/6 inhibitor's action requires Rb, sadly, the CDK4/6 inhibitor has not been used in TNBC patients and Rb-loss patients have been excluded from CDK4/6 inhibitor trials. For reviewer's easy reference. We have systematically curated a list of all current clinical trials using CDK4/6 inhibitor from *Clinicaltrial.gov* as Table 1 below. Here we want to reiterate that the key novelty of our study, as acknowledged by both reviewer 1 and reviewer 2, is our mechanistic study provide a preclinical rationale that CDK4/6 inhibitor

can be used in TNBC breast cancer patients, regardless of the tumor's Rb status. Importantly, our finding provides a clear preclinical rationale for combinatorial treatment, which would allow an extension of current breast cancer CDK4/6 inhibitor clinical trials (as listed in the Table1) to previously excluded TNBC patients. The future clinical proposition of the CDK4/6 inhibitor-containing regimen to TNBC is a significant advance for TNBC treatment.

Table 1 Current Clinical Trials Using CDK4/6 inhibitors

Cancer Type	Phase	Biomarkers	Combo regimen	Sponsor/location	Year
Breast Cancer	II	HER2+	Palbociclib + Trastuzumab	Ciruelos- SOLTI Breast Cancer Research Group	In process, started May 2015
Breast Cancer	II	ER+ HER2+	Abemaciclib + Trastuzumab	Eli Lilly and Company	In process, started February 2016
Breast Cancer	II	HER2+	Palbociclib + Trastuzumab	Santa-Maria – Northwestern University	In process, started May 2016
Breast Cancer	Ib/II	HER2+	Ribociclib + Trastuzumab Ribociclib + T-DM1	Tolaney- Dana-Farber Cancer Institute	In process, started January 2016
Breast Cancer	Ib	HER2+ Rb+ Low p16in4a	Palbociclib + T-DM1	Haley-University of Texas Southwestern Medical Center	In process, started November 2013
Breast Cancer	I/II	ER+ HER2+	Palbociclib + T-DM1 + Letrozole	Nye- University of Kansas Cancer Center	In process, started October 2018
Breast Cancer	II	HER2+	Palbociclib + T-DM1	Chalasani- University of Arizona	In process, started May 2018
Breast Cancer	I	ER+ HER2+	Abemaciclib + Trastuzumab Abemaciclib + Trastuzumab + Pertuzumab	Eli Lilly and Company	In process, started February 2014
Breast Cancer	III	ER+ HER2+	Palbociclib + Trastuzumab + Endocrine Therapy Palbociclib + Pertuzumab + Endocrine Therapy	Metzger- Alliance Foundation Trials, LLC	In process, started October 2016
Breast Cancer	II	ER+ HER2+	Palbociclib + Trastuzumab + Pertuzumab + Letrozole	International Breast Cancer Study Group	In process, started August 2018
Breast Cancer	II	ER+ HER2+	Palbociclib + Trastuzumab + Pertuzumab	Gianni-Fondazione Michelangelo	In process, started August 2015
Breast Cancer	I/II	HER2+	Palbociclib + Trastuzumab + Pertuzumab + Anastrozole	Tiersten- Icahn School of Medicine at Mount Sinai	In process, started October 2017
Breast Cancer	II	ER+ HER2+	Palbociclib+ Trastuzumab+ Letrozole	Ademuyiwa- Washington University School of Medicine	In process, started September 2016
Non-small	I/II	EGFR+	G1T38 + Osimertinib	G1 Therapeutics	In process,

cell lung cancer					started March 2018
Non-small cell lung cancer	Ib	EGFR+	Ribociclib + EGF816	Novartis Pharmaceuticals	In process, started November 2017
HNSCC	I/II		Palbociclib + Cetuximab	Adkins- Washington University School of Medicine	In process, started April 2014
HNSCC	II		Palbociclib + Cetuximab	Pfizer	In process, started July 2015
HNSCC	I		Ribociclib + Cetuximab	Cliniques universitaires Saint-Luc- Université Catholique de Louvain	In process, started April 2015
HNSCC	I/II		Palbociclib + Cetuximab + IMRT	Ngamphaiboon- Mahidol University	In process, started January 2017
HNSCC	I		Palbociclib + Avelumab + Cetuximab	Gold- UC San Diego	In process, started April 2018
Solid Tumors	I	EGFR+ HER2+	Palbociclib + Neratinib	MD Anderson Cancer Center	In process, started February 2017
Non-small cell lung cancer	I/II	KRAS mutant	Palbociclib + PD-0325901	Shapiro- Dana-Farber Cancer Institute	In process, started December 2013
Pancreatic Cancer	Ib/II	KRAS-mutant (Colorectal)	Ribociclib + Trametinib	Novartis Pharmaceuticals	In process, started March 2016
Colorectal Cancer					
Solid Tumors	I		Palbociclib + Trametinib	GlaxoSmithKline	Completed June 2016
CNS tumors	I		Ribociclib + Trametinib	St. Jude Children's Research Hospital	In process, started February 2018
Non-small Cell Lung Cancer	I/II	KRAS-mutant	Palbociclib + Binimetinib	Shapiro- Dana-Farber Cancer Institute	In process, started May 2017
Melanoma	Ib/II	NRAS mutant	Ribociclib + Binimetinib	Array BioPharma	Completed February 2018

2. The use of DEDD expression as potential biomarker for the combination of Abemaciclib and Lapatinib in TNBC is questionable as cell lines with low expression are also sensitive to the combination (HCC1806, KBT-21, KBT-37; Figure 5E,6B). This leaves doubt about the dependency on DEDD overexpression and the potential therapeutic window for this combination.

Answer:

We agree that cell lines with normal expression of DEDD or moderate upregulation of DEDD responded to Abemaciclib and Lapatinib (comb) treatment in some degree as the reviewer noted. In regard to reviewer's concern of "therapeutic window", we do believe 1) DEDD expression level directly regulates the overall sensitivity of a given cell to combo treatment, and 2) overexpression of DEDD in tumor cells provide a valuable therapeutic window distinct from normal cells. The following evidence supports our conclusion:

First, under the same combination treatment dosage (new Fig. 5a), TNBC cell lines with higher DEDD expression, e.g. HCC38, MDA-MB-468, BT-20 and HCC1937 (please refer to Fig. 2a for DEDD expression western blot), showed a higher sensitivity to the combo treatment than those with a moderate or lower DEDD expression (HCC1806, MDA-MB-436 and MDA-MB-231) (Fig. 5a). Such a strong correlation between DEDD expression and sensitivity to combo treatment provided the first line of evidence, suggesting a greater dependency of DEDD in TNBCs in response to the combo treatment.

Second, beyond above correlative observation, to further demonstrated the functional dependency of DEDD expression pertains to the sensitivity to the CDK4/6 inhibitor-containing regimen, we modulated the DEDD expression in both normal mammary tissue cell lines (KTBs, DEDD low expression) as well as TNBC cells (MDA-MB-436, DEDD moderate expression). As shown in Fig. 5d-g, ectopic expression of cytosol DEDD in KTB cells or overexpression of cytosol DEDD in MDA-MB-436 cells rendered above cells more sensitive to both Abemaciclib treatment alone and combo treatment. Such increased sensitivity to CDK4/6 inhibitor regimen, as a direct result of DEDD overexpression, indicated a causal role of DEDD level in shaping the overall sensitivity of given cells to the combo treatment.

Lastly, considering the high frequency of genetic amplification and upregulation of DEDD mRNA in a large portion of TNBC clinical samples (67% in ER-/PR-/HER2 normal tumors, cBio TCGA dataset, updated Fig. 1c) and in TNBC cell lines (Fig. 2a and Supplementary Fig. 6d), we believe TNBC patients with significant DEDD dysregulation will mostly likely be more sensitive to combo treatment than tumors with lower or normal DEDD expression.

We fully appreciate the reviewer's concern on our conclusion. An affirmative conclusion of the clinical validity of DEDD as a biomarker for the combo treatment will rely on future retrospective/prospective clinical studies in TNBC patients. Therefore, in this revised manuscript, we have tuned down our conclusion regarding the clinical implications of DEDD as a biomarker for combo treatment by removing the previous discussions regarding the biomarker (see revised manuscript Discussion section, page 22. line 3-7).

3. In addition, the work does not provide a mechanistic explanation for the observed synergy between the two treatments, why does abemaciclib alone increase pERK and does it reduce pAKT in the presence of lapatinib (Figure 5C)?.

Answer:

We thank the reviewer for this important point. In this revised manuscript, we conducted additional analysis and revealed one of the potential mechanisms that contribute to observed synergistic inhibition of AKT activation.

From the time course analysis of lapatinib treatment (Supplementary Figure 1c, Supplementary Figure 8c), we noticed that lapatinib transiently suppressed p-AKT in multiple TNBC cell lines, followed by a regaining p-AKT 24 hours post-treatment (Supplementary Figure 1c, Supplementary Figure 8c). Such regaining of p-AKT is dependent on DEDD, as knocking down DEDD reduces p-AKT at 24 hrs post-treatment (new Supplementary Figure 3F).

The increased both AKT protein level and phosphorylation level at 24 hours (a delayed response) might also suggest the existence of downstream compensatory mechanism(s) that lead to reactivation of AKT, most likely through transcriptional regulation rather than protein level regulation. We postulated that such mechanisms might be regulated through CDK4/6 because of observed synergy between lapatinib and CDK4/6 inhibitor. Literature search led us to hypothesize that transcription activation FoxM1 might be a downstream compensatory mechanism in our context. FoxM1 has been reported to be a CDK4/6 downstream substrate³ and the activation of FoxM1 forms a positive feedback loop to the AKT activation⁴.

Indeed, after abemaciclib treatment, we observed a significant decrease in both total FoxM1 and p-FoxM1 level in TNBC cells (updated Fig. 5c). This new piece of evidence suggests, when p-Akt is inhibited by lapatinib and FoxM1-AKT positive feedback is engaged, blocking CDK4/6 leads to loss of FoxM1 activation, and a subsequent reduction of p-AKT. We have updated our model (new Figure 7) to reflect this new observation.

As for the reviewer's notion that "abemaciclib treatment alone increase p-ERK", similar observation has been reported before in other tumor context⁵. This observed p-ERK was not accompanied by an increase in functional downstream signaling such as p-mTOR. Therefore, the previous report concluded that p-ERK is most likely mediated by acute changes in protein phosphorylation⁵. While teasing out the intricate negative feedback loop of ERK signaling by abemaciclib is beyond the scope the current manuscript, our data did show that combining lapatinib can effectively reduce the abemaciclib-induced p-ERK increase (Fig. 5c,e and Supplementary Fig. 8c), which partially explains the combinatorial efficacy.

4. The authors have studied the effects of knock-down, over-expression and inhibitor treatment in many different cell lines and models. However, in many instances, observations are first made in one cell line and subsequently further explored in other cell lines. This makes it difficult to interpret the data and draw strong conclusions. This is exemplified by Figure 3H, 3J and K, and the absence of the PDX HCI-004.

Answer:

In this revised manuscript, we performed an additional co-IP experiment to study cytosol DEDD-HSC70 interaction in non-cancer cells 293FT and observed a similar interaction pattern between cytosol DEDD and HSC70. The new data have been incorporated into new Fig. 3g. Fig. 3h, 3i and 3j have also been updated in the revised manuscript. For Fig. 3h, a representative FACS of KTB-37 has been shown. For Fig. 3i, representative western blots of KTB-34 and KTB-37 cell lines have been shown. And for Fig. 3j, now we have included KTB-21, KTB-34 and KTB-37 data to show the consistent conclusions. As for the *in vivo* experiment, we have already provided extensive data to demonstrate the generality of the combinatorial efficacy by using three *in vivo* models (HCC1806, MDA-MD-468 and TNBC PDX HCI-001). Adding TNBC HCI-004 PDX *in vivo* experiment will unlikely to provide additional significant mechanistic insights to our current conclusion.

5. This inconsistency is also observed in different Figures testing the same perturbation or condition, for example, the lack of an increase in RB levels by knockdown of DEDD in Figure 4C compared to Figure 4A. Also, why is the overexpression of FLAG-dNLS-DEDD not downregulating RB expression (in the 1% input lane) in Figure 4F compared to Figure 4D.

Answer:

We would like to clarify that Fig. 4A and 4C have very different experimental settings and distinct goals. Thus, the western blot results from the above two figures are not directly comparable. For Fig. 4A, the goal of this experiment is to examine the impact on shDEDD on Rb expression. Cells are collected under regular cell culture condition. For Fig. 4C, the goal of this experiment is to examine the time-dependent accumulation of Rb protein after blocking protein degradation pathway. The cells are serum starved for 12 hours first to synchronize the cell cycle. Then fresh FBS-containing media and MG-132 were added (time 0) prevent proteasomal-mediated degradation. For Fig. 4F, in order to reliably detect Rb-DEDD interaction, the Rb degradation (induced by DEDD) needs to be prevented. We added MG-132 throughout the cell culture. Therefore, it is not surprising that we did not see the Rb protein level decrease. In the case of Fig. 4D, the MG-132 was not added, except for the last four-lane. When MG-132 was added, we did not see Rb level decrease, which is consistent with Fig. 4f.

6. A major concern with thus work that most of the observations are based on either knock-down using shRNA or over-expression of DEDD or DEDD mutants. Especially there is concern about the reproducibility of the effects of shRNA knock-down with other means of perturbation. The authors should take caution in using high titer lentiviruses encoding shRNAs and preferably perform the experiments at low MOI to ensure single integrations.

Answer:

We completely agree with the reviewer on this point. As a matter of fact, all of our lentiviral experiments have been conducted by using a lentiviral stock at a titer of MOI 0.6 or below for shRNA knockdown or overexpression. The backbone lentiviral plasmids (pLKO and pLenti-CMV-Puro) used to have a puromycin Puromycine (1-5 ug/ml) selection after 48 hour viral infection showed less than 50% puromycin resistant cell in any given lentiviral experiments. For the initial shRNA screening, we used an MOI ~ 0.3-0.6. The MOI chart used for our shRNA screen is included here.

7. In addition, the authors should validate the results with CRISPR-based inactivation (e.g. CRISPRi or a conditional intron and CRE recombinase). It is disconcerting that the large-scale efforts across many cell lines for dependencies do not show an effect of loss of DEDD expression.

Answer:

We are very much aware of the large-scale efforts across many cell lines for gene dependencies (depmap.org). The existing CRISPR screen data (CRISPR (Avana) Public 19Q1) showed a moderate dependency score of DEDD across multiple cell lines. **Most importantly**, all the data curated at depmap.org are based on a given gene's straight lethal effect, but not a synthetical lethal activity under EGFR/HER2 inhibition. In our shRNA screening design, we first excluded straight lethal genes after shRNA library infection and puromycin selection. Then we exclusively focused on genes that have synthetic lethal interactions with EGFR signaling inhibition (see Fig.1a). Therefore, there are no direct comparison can be conducted between our data and existing screening data, as loss of DEDD is not straight lethal.

8. In addition there seems to be little correlation with knock-down and the effect-size. For example, in Figure S3a, there seems a lack of correlation with knock-down, shABCB10-5 best enhancer with the least knock-down?

Answer:

The biological variation of the effects of different shRNA is expected. Therefore, one of the very important steps of shRNA screening experiment is to independently validate the screening hits before the further mechanistic study. When we conducted independent shRNA validation using multiple TNBC cell lines, we didn't observe a consistent pattern that knocking down of ABCB10 or UAP1 re-sensitize TNBC cell lines to the lapatinib treatment (Supplementary Figure 3b, c, d and e). Although all three shRNAs effectively knockdown ABCB10 or UAP1 in two TNBC cell lines, only two shRNAs can resensitize HCC1806 cells to the lapatinib treatment, while none of them show significant effects on MDA-MB-468 cells (Supplementary Figure 3c, e). Compared to ABCB10 and UAP1, knockdown of DEDD showed consistent synergistic effects in all three TNBC cell lines (Fig. 1F). Therefore, we chose DEDD as our focus for this manuscript.

9. In Figure S3b, there is a very small effect of complete knock-down and in Figure S3c and Figure 2F, only 2 shRNAs were tested for effect, whereas for all others 5 shRNA were used?

Answer:

Among five ABCB10 shRNAs do not knockdown ABCB10 very efficiently in both HCC1806 and MDA-MB-468 cells. We, therefore, chose the three most effective ABCB10 knockdown constructs to create stable ABCB10 knockdown TNBC cells and performed cell viability using those stable cell lines (updated Supplementary Fig. 3b, c). We applied similar criteria by selecting the most effective DEDD-targeting shRNA constructs. In Fig. 1F, the three most effective DEDD shRNAs constructs were tested

for synergistic effect under lapatinib treatment. To avoid confusion, in this revised manuscript we have updated Fig. S3 by only presenting three most effective shRNAs.

10. Also in Figure S3C, protein expression should be provided in addition to mRNA expression. This in relation to Figure 4A and B describing dependency upon knock-down.

Answer:

DEDD protein expression levels of DEDD shRNA knockdown TNBC cell lines have been provided in updated Fig. 4a and h.

11. The basis of this work is an shRNA screen but the details of the screen and the interpretation of the screen is insufficient and not presented in a proper way. The list is an alphabetical sorted list of genes. The authors should at least provide raw and processed data of the individual replicates, individual shRNA depletion scores, and some sort of method to score the behavior of shRNAs targeting the same gene (e.g. RRA or MAGECK). Finally, this list should provide fold depletion in treated and untreated conditions accompanied by statistical significance.

Answer:

We thank you for the reviewer's suggestion. We reanalyzed our shRNA screening raw data using MAGECK software package as suggested⁶. The full results are now included in the revised manuscript as Supplementary Table 1 and Supplementary Figure 2.

12. Figure 1f, normalized cell to the number to what, DMSO control for each shRNA? Cannot exclude the strong effect on untreated cells (as for MDA-MB-468 in Figure 2C).

Answer:

The cell viability of lapatinib treated cells were normalized to their untreated groups (DMSO) in each shRNA. Thus, we are comparing cell viability between the lapatinib treatment group and DMSO control group within the same stable DEDD shRNA knockdown cell lines (Fig. 1F). For Fig. 2c, we are comparing cell viability between the stable DEDD shRNA knockdown cell lines and the pLKO.1 control group. We have clarified this point in the revised Figure legends.

13. What means the annotation of the axis in Figure 3E; 100%?

Answer:

In this revision, we have changed the label to "EdU Incorporation (%)" in new Figure 3D in the version of the manuscript.

14. Supplemental Figure 4B what is the FDR cut-off?

Answer:

As shown in the updated Supplementary Figure 4a, FDR cut-off is 0.012.

15. How were the correlations to DEDD levels in Figure 2E determined? Quantitative? Normalized to beta-Actin (seems overexposed and not quantitative?)

Answer:

As suggested by the reviewer, we performed quantification by normalizing western blot band density to beta-actin level and quantified DEDD expression levels with different cyclin expression levels at different time points in multiple TNBC cell lines (see new Supplementary Fig. 5d). In HCC1806 and HCC38 cells, the expression dynamic pattern of DEDD is similar to cyclin D1, but not other cyclins, suggesting a correlation expression between DEDD and cyclin D1. To examine the causal role of DEDD expression on different cyclins, we performed additional siRNA DEDD knockdown experiments in both HCC38 and HCC1806 cells in addition to MDA-MB-468. As shown in the new Fig. 2f, knocking down DEDD strongly reduced cyclin D1 level and slightly reduced cyclin B1 expression.

16. Evidence for cytoplasmic localization insufficient. The authors should show the analysis of TNBC cells lines with different levels of DEDD for the localization by both IHC and cell fractionation followed by western blotting.

Answer:

To examine the DEDD subcellular localization, we performed IF staining of DEDD in multiple TNBC cell lines (new Supplementary Figure 6a). While DEDD expression showed a heterogeneous pattern across multiple TNBC cell lines, DEDD is enriched in the cytosol compartment in all three TNBC cell lines tested.

17. *The effects of overexpression of DEDD or dNLS-DEDD are the same on cell cycle progression. How can the authors conclude that cytoplasmic localization of DEDD is responsible for the observed phenotype?*

Answer:

DEDD is localized in both cytosol and nucleus. To address the DEDD's cytosol specific function, we employed a Δ NLS-FLAG-DEDD expression plasmid, which is not capable entering into the nucleus (Supplementary Fig. 6b) due to the loss-of-function mutations on all three nuclear localization sequences as reported before⁷. To address the DEDD nuclear specific function, in this revised manuscript, we first treated 293FT cells with nuclear export inhibitor leptomycin B (Lepto). Treatment of leptomycin B significantly reduced the cytosol fraction of wt-DEDD (new Fig. 3b). Under this condition, the increased S phase cell population caused by wt-DEDD overexpression reduced significantly (Fig. 3b), indicating that cytosolic expression of DEDD is the major contributor to G1-S transition. Taken together, cytosol DEDD is the major contributor for the DEDD's cell cycle promoting effects.

18. *In Figure 3J levels of DEDD are strongly reduced by siHSC70 and this could readily explain the reduction in S-phase increase in Figure 3i.*

Answer:

We completely agree the reviewer's point. Indeed, the DEDD protein level appears to be dependent on HSC70.

19. *The authors should provide the expression levels of dNLS-DEDD in KBT-21 for Figure 5E as this can be very low as can be seen in Figure 4E, no MG-132.*

Answer:

We agreed with the reviewer's point. The expression levels of Δ NLS-DEDD in KBT-21 and KTB-37 cells were provided side by side with the cell viability test in the updated Figure 5D.

20. *What about Xenograft HCI-004? The authors should provide growth curves of both PDX models in absence and presence of treatment similar to Figure 6E.*

Answer:

HCI-004 has a very slow growth rate *in vivo*. In our hand, it normally takes 6 months for the transplanted the PDX tumors to be palpable. Therefore, we decided it is not practical to perform xenograft experiment using HCI-004. However, with our data of two TNBC cell line HCC1806 and MDA-MB-468 xenografts and one PDX xenografts, we believe we have provided compelling evidence that TNBCs with an elevated DEDD expression (Fig. 6b, c), and the combination treatment of combining both anti-EGFR and CDK4/6 inhibitor significantly suppresses TNBC tumor growth *in vivo* (Figure 6d, e, f, g, h, I, Supplementary Figure 9a, b and c).

References cited in this document:

1. Cerami, E. *et al.* The cBio Cancer Genomics Portal: An Open Platform for Exploring Multidimensional Cancer Genomics Data. *Cancer Discov.* **2**, 401–404 (2012).
2. Gao, J. *et al.* Integrative Analysis of Complex Cancer Genomics and Clinical Profiles Using the cBioPortal. *Sci Signal* **6**, p11–p11 (2013).
3. Anders, L. *et al.* A Systematic Screen for CDK4/6 Substrates Links FOXM1 Phosphorylation to Senescence Suppression in Cancer Cells. *Cancer Cell* **20**, 620–634 (2011).

4. Yu, G. *et al.* FoxM1 promotes breast tumorigenesis by activating PDGF-A and forming a positive feedback loop with the PDGF/AKT signaling pathway. *Oncotarget* **6**, 11281–11294 (2015).
5. Goel, S. *et al.* Overcoming Therapeutic Resistance in HER2-Positive Breast Cancers with CDK4/6 Inhibitors. *Cancer Cell* **29**, 255–269 (2016).
6. Li, W. *et al.* MAGeCK enables robust identification of essential genes from genome-scale CRISPR/Cas9 knockout screens. *Genome Biol.* **15**, (2014).
7. Stegh, A. H. DEDD, a novel death effector domain-containing protein, targeted to the nucleolus. *EMBO J.* **17**, 5974–5986 (1998).

REVIEWERS' COMMENTS:

Reviewer #1 (Remarks to the Author):

The authors responded to most of my concerns adequately. The reviewer requests no more revision.

Reviewer #2 (Remarks to the Author):

I have no further comments. The authors addressed my questions appropriately.

Reviewer #3 (Remarks to the Author):

The authors have strengthened this work with additional experiments, controls and textual changes and additions to clarify their work. I agree with the other reviewers that this work is of interest describing a potential combination treatment for TNBC.

The authors have now included the raw data and analysis of the screen. However, they do not draw any conclusions on the interpretation of the data. I would like to point out that the interpretation of the pooled shRNA screen is quite optimistic. None of the selected genes (top 200 MAGeCK) are significant with respect to an $FDR < 0.1$. The authors have selected 4 genes based on their genomic alterations (cut-off of $>40\%$?) in TNBC but one wonders how this analysis would come out with 200 other genes randomly chosen from the gene list, or how a ranked list of a full genome genomic analysis of TNBC would look like. Of the four selected genes, only DEDD validates in HCC1806 cells. Of note, the validation in two other TNBC cell lines is much stronger, prompting the authors to pursue DEDD as the only candidate for follow-up.

The experiments addressing the biological function of DEDD in the response to Lapatinib are convincing and do suggest an additional role for DEDD in the control of different cell cycle proteins. Of note, although the authors mention that the screen filtered against straight lethal genes, the knockdown of DEDD in HC38 and MDA-MB-468, reduced cell number with more than 70%. This makes the interpretation of figure 1F more challenging because the effect was normalized against each individual shRNA rather than a common reference.

In conclusion, the experiments described in this manuscript support a novel cytoplasmic role for DEDD in the control of cell cycle proteins. This model has led to the potential combination of CDK4/6 and Lapatinib in TNBC which was validated in xenograft experiments. However, I do struggle with the selection of DEDD as interesting hit from the screen and would like to avoid the impression that the interpretation without any consideration for significance, is exemplary for large scale screening.

Point-by-Point Response to Reviewers' Comments

Re: Manuscript # NCOMMS-18-32121A-Z

We would like to thank the editor and the reviewers for their valuable and constructive comments and for consideration of our revised manuscript to be published on *Nature Communications*. We have carefully considered Reviewer 3's comments and address reviewers' comments in full by changing both the title and the main text. Here, we have indicated the changes made to the manuscript to take into account the comments from the Reviewer 3, which are reproduced below in *italics* (font size 10) for your easy reference. Our point-by-point replies to every comment from the reviewers are as follows:

Reviewer #3:

1. *The authors have now included the raw data and analysis of the screen. However, they do not draw any conclusions on the interpretation of the data. I would like to point out that the interpretation of the pooled shRNA screen is quite optimistic. None of the selected genes (top 200 MAGeCK) are significant with respect to an FDR<0.1.*

Answer: We thank the reviewer for this comment on the interpretation of the RNAi screen hit selection criteria. Due to the intrinsic noisy nature of whole genome RNAi screening, we agree with the reviewer that the FDR value in our screen is not ideal as compared to commonly seen in CRISPR-based screen. However, in our case, the goal of the preliminary RNAi screen was not to define a gene function, rather generate a list of genes with lower p-value/FDR that are clinically relevant to TNBC (Fig. 1C-E, TCGA analysis) for further hypothesis testing. With this consideration, we selected the top 200 genes in the MAGeCK **ranked** list. The value of the RNAi screen was to enable us to discover the commonly amplified Chromosome 1q cluster genes, including DEDD. Most importantly, through independent shRNA validation and further mechanistic study (Fig. 2-6), we demonstrated DEDD is a synthetic lethal gene under Lapatinib (LAP) treatment. Since the functional synthetic lethal screen is not the major goal of our manuscript and did not encompass the majority of the findings presented, we agree with the reviewer that our original title, "*Functional Screen Identifies Triple-Negative Breast Cancer Vulnerability Driven by Death Effector Domain-Containing Protein*", did not fully capture the theme of our study. Thus, in this final revision, we have changed both the title and the main text to minimize the impression of "...exemplary for large scale screening" (as reviewer stated in comment 5). Instead, we put more emphasis on the functional role of DEDD in regulating cell cycle and its significant role in affecting the combinatorial treatment of CDK4/6 inhibitor and the EGFR inhibitor. The new title now reads as: "*Death Effector Domain-Containing Protein Renders Vulnerability to Cell Cycle Inhibition in Triple-Negative Breast Cancer*".

2. *The authors have selected 4 genes based on their genomic alterations (cut-off of >40%?) in TNBC but one wonders how this analysis would come out with 200 other genes randomly chosen from the gene list, or how a ranked list of a full genome genomic analysis of TNBC would look like.*

Answer: We thank the reviewer for this comment. Indeed, the top 200 genes are Top 200 "**negrank**" list selected from MAGeCK analysis data (see Supplementary Data 1, tab "MAGeCK_gene_summary"). As explained in our answer to reviewer's comment 1, to explore the clinical relevance of these genes in TNBC context, we overlaid these 200 genes with clinical amplification data from cBioPortal, adding the clinical significance of those potential hits.

3. *Of the four selected genes, only DEDD validates in HCC1806 cells. Of note, the validation in two other TNBC cell lines is much stronger, prompting the authors to pursue DEDD as the only candidate for follow-up.*

Answer: We appreciate that reviewer reiterated our hit selection rationale. We did not intend to claim these 4 genes are the definite signature for TNBC. As clinical TNBCs are highly heterogeneous, these genes might have different significance in certain TNBC tumor. Since we could not validate DEDD's effect in all three TNBC cell lines, DEDD appeared to be a more common alteration in TNBC. Thus, we pursued DEDD in the following experiments and studied DEDD's role in TNBC's sensitivity to cell cycle inhibition.

4. *Of note, although the authors mention that the screen filtered against straight lethal genes, the knockdown of DEDD in HC38 and MDA-MB-468, reduced cell number with more than 70%. This makes the interpretation of figure 1F more challenging because the effect was normalized against each individual shRNA rather than a common reference.*

Answer: We thank the reviewer for this comment. Compared with HCC1806, both HCC38 and MDA-MB-468 have higher expression of endogenous DEDD (Fig. 2A). In fact, the result of both cell lines are more sensitive DEDD knockdown-induced straight lethal (Fig. 2B) supports the notion that DEDD amplification creates a TNBC vulnerability in a dose-dependent manner. Because of this significant difference of knocking down DEDD on each TNBC cell line, it makes challenging to interpret data when we study the Lapatinib's treatment effect across different cell lines (Fig. 1F). Therefore, we chose to normalize to DMSO treatment within each TNBC cell line and shRNA-sublines and consider the baseline viability as 100% for each cell line. By comparing each cell line's own DMSO baseline, we could more clearly observe the Lapatinib's effects. As shown in Fig. 1F, the difference between Lapatinib and DMSO treatment are much more significant than the differences in the pLKO.1 control group. Thus, in our study, in addition to DEDD's cell cycle regulation functional in TNBCs, we do have evidence supporting DEDD also contributes to the resistance to anti-EGFR treatments (Fig. 1F, Fig. 5 and Fig. 6,7). We hope our answers will clarify the reviewer's concerns.

5. *However, I do struggle with the selection of DEDD as interesting hit from the screen and would like to avoid the impression that the interpretation without any consideration for significance, is exemplary for large scale screening.*

Answer: We agree with the reviewer and appreciate the expert's perspective. As we have explained in our answer to reviewer's comment 1, in the revised manuscript, we removed the phrase "functional screen" from the title, and the main text has also been adjusted to tune down the emphasis on the functional screen (see the manuscript main text results section **Page 6**). Our study did provide evidence that DEDD is contributing to the resistance to the anti-EGFR treatment and have strong applications in TNBC's sensitivity to CDK4/6 inhibitors containing combinational treatments. In addition, we revealed novel mechanisms by which DEDD regulates the G1-S cell cycle transition. We believe the updated title, "*Death Effector Domain-Containing Protein Renders Vulnerability to Cell Cycle Inhibition in Triple-Negative Breast Cancer*", more appropriately reflects the findings presented in this manuscript.